# Online Learning in the Random-Order Model

**Martino Bernasconi** [* 1]   **Andrea Celli** [* 1]   **Riccardo Colini Baldeschi** [* 2]   **Federico Fusco** [* 3]   **Stefano Leonardi** [* 3]   **Matteo Russo** [* 3]

## Abstract

In the random-order model for online learning, the sequence of losses is chosen upfront by an adversary and presented to the learner after a random permutation. Any random-order input is *asymptotically* equivalent to a stochastic i.i.d. one, but, for finite times, it may exhibit significant *non-stationarity*, which can hinder the performance of stochastic learning algorithms. While algorithms for adversarial inputs naturally maintain their regret guarantees in random order, simple no-regret algorithms exist for the stochastic model that fail against random-order instances.

In this paper, we propose a general template to adapt stochastic learning algorithms to the random-order model without substantially affecting their regret guarantees. This allows us to recover improved regret bounds for prediction with delays, online learning with constraints, and bandits with switching costs. Finally, we investigate online classification and prove that, in random order, learnability is characterized by the VC dimension rather than the Littlestone dimension, thus providing a further separation from the general adversarial model.

## 1. Introduction

Random order is a natural input model for online algorithms: the input is generated upfront by an adversary but is presented to the algorithm after a uniform random permutation. The random-order model is extensively studied across various areas of computer science and economics, such as optimal stopping problems (Dynkin, 1963; Krengel & Sucheston, 1977; Peng & Tang, 2022), online matching and matroid selection (Babaioff et al., 2018; Korula & Pál, 2009; Dimitrov & Plaxton, 2012; Ezra et al., 2022), and online approximation algorithms (Gupta & Singla, 2020). However, it has received little attention in online learning, where the standard input regimes are either (i.i.d.) stochastic or adversarial (see, e.g., Lattimore & Szepesvári, 2020). The study of online learning in the random-order model is part of a broader research agenda that explores meaningful models beyond worst-case analysis (Roughgarden, 2020). Examples of other "intermediate" input models for online learning can be found in Ben-David et al. (1997); Slivkins & Upfal (2008); Lykouris et al. (2018); Sachs et al. (2022); Haghtalab et al. (2022); Mazzetto & Upfal (2023).

In random-order online learning, the $T$ loss vectors are generated arbitrarily by some adversary and then undergo a uniform shuffling before being presented to the learner. This input model is — to some extent — intermediate between the stochastic model, where losses are drawn i.i.d. from a fixed but unknown distribution, and the adversarial one, where the adversary decides both the losses *and* the order in which they are presented to the learner.

A classical probabilistic result by De Finetti (1929) (see also De Finetti, 1937; Hewitt & Savage, 1955; Diaconis & Freedman, 1980) on sequences of exchangeable random variables implies that sampling without replacement (i.e., the distribution over random-order loss sequences) is *asymptotically* indistinguishable from i.i.d. sampling from some fixed distribution over losses[1]. However, in the standard *finite-time* online learning framework, the non-stationarity of the random-order model can undermine learning algorithms that perform well in the stochastic setting. This observation raises several questions about the random-order model and its relationship with other input models.

### 1.1. Our Results

In this paper, we initiate the systematic study of the random-order input model in online learning. Since any realization of random-order sequences of losses can be naturally captured by the standard adversarial model, it is immediate to see that any algorithm for the adversarial model maintains its guarantees in random order. Conversely, any i.i.d. se-

---

[*]Equal contribution  [1]Department of Computing Sciences, Bocconi University, Milan, Italy [2]Central Applied Science, Meta, London, UK [3]Department of Computer, Control and Management Engineering "Antonio Ruberti", Sapienza University, Rome, Italy. Correspondence to: Federico Fusco <fuscof@diag.uniroma1.it>.

*Proceedings of the 42^{nd} International Conference on Machine Learning*, Vancouver, Canada. PMLR 267, 2025. Copyright 2025 by the author(s).

---

[1]For more details, we refer to Appendix A.

quence can be simulated by a random-order one by first sampling $T$ losses and then applying a uniform permutation. These two considerations show that the random-order model lies between the i.i.d. and adversarial input model[2]:

$$i.i.d. \preceq random\text{-}order \preceq adversarial.$$

The hierarchy between the three models implies that investigating the random-order model is especially compelling in problems that exhibit a performance gap between the stochastic and adversarial models. In such problems, a natural question is: *What happens when we apply an algorithm designed for the stochastic setting to the random-order model?* One might expect such algorithms to perform reasonably well since the adversary can only select the support of the distribution. However, we show an algorithm that behaves nicely against stochastic inputs, but fails in a carefully designed random-order instance (Section 3). We complement this negative result with a positive one. While we cannot directly apply an algorithm designed for the stochastic setting to the random-order model, we propose a general template, SIMULATION, to adapt learning algorithms from the i.i.d. setting to the random-order model, without significantly affecting their regret guarantees (Section 4). This implies that the minimax regret regimes in the random-order model essentially *collapse* to stochastic i.i.d. ones. We demonstrate the applicability of our template across multiple problems that exhibit a stark separation between the adversarial and stochastic regimes.

**Prediction with Delayed Feedback.** The interplay between online learning and delayed feedback has been studied extensively. While many delay models exist, the general finding is that the delay parameter $d$ influences regret bounds *additively* in the i.i.d. model and *multiplicatively* in the adversarial one (Desautels et al., 2012; Joulani et al., 2013; Masoudian et al., 2022). In Corollary 4.4, we show an algorithm with $\widetilde{O}(\sqrt{T} + d)$ regret rate in the random-order model, which qualitatively matches the stochastic result.

**Online Learning with Constraints.** In online learning problems with time-varying constraints there is a strong separation between the adversarial case, where it is impossible to attain sublinear regret and constraints violations (Mannor et al., 2009), and the stochastic case, where $\widetilde{O}(\sqrt{T})$ regret is achievable (Yu et al., 2017). We present an algorithm (Corollary 4.8), based on SIMULATION, that guarantees regret of order $\widetilde{O}(\sqrt{T})$ and zero constraints violation, matching the results for stochastic environments.

**Bandits with Switching Costs.** In the bandits with switching costs problem, the learning algorithm incurs an additional unitary loss every time it changes action. The adver-

sarial minimax regret is $\Theta(T^{2/3})$, while the stochastic one is $\Theta(\sqrt{T})$ (Cesa-Bianchi et al., 2013; Dekel et al., 2014). We explore a simplified version of SIMULATION based on SUCCESSIVE-ELIMINATION, similar to the algorithm by Agrawal et al. (1988), and show that this approach recovers the $\sqrt{T}$ rate in the random-order model.

**Online Classification.** For (binary) classification, there is a well-known separation between statistical (offline) learning[3] and online learning. While the learnability in the former setting is characterized by the Vapnik–Chervonenkis (VC) dimension, the latter is determined by the Littlestone dimension (Ben-David et al., 2009). In general, the Littlestone dimension dominates the VC dimension, and there are simple examples (e.g., one-dimensional thresholds) of families with constant VC dimension and infinite Littlestone dimension. In Appendix H, we prove that the VC dimension characterizes learnability also in the random-order model, thus providing a further separation between this model and the general adversarial one. Notably, results similar in spirit to ours have been recently obtained in Haghtalab et al. (2022); Raman & Tewari (2024), where they prove that online binary classification is characterized by the VC dimension *even* in a non-stationary setting, as long as the adversarial input is smooth (Haghtalab et al., 2022) or the learner has access to good predictions (Raman & Tewari, 2024).

**Discussion.** On a theoretical level, our results have two main implications: (i) the minimax regret rates for random-order and the stochastic model are typically the same, and (ii) it is fairly easy to construct the desired random-order algorithm starting from a stochastic one. These results also have important practical implications: if the order in which the input is presented can be controlled, the same regret rate achievable in the stochastic setting can be obtained by simply shuffling the dataset.

### 1.2. Our Techniques

**The Birthday Paradox.** We show that algorithms designed for the stochastic setting may fail in the random-order model, by leveraging the birthday paradox (from a pool of $n$ elements, a duplicate appears after about $\sqrt{n}$ samples). In a pool of $T$ loss vectors, a random-order instance corresponds to uniform sampling without replacement, whereas sampling with replacement mimics an i.i.d. distribution. When the support is finite, these processes are statistically distinct (Diaconis & Freedman, 1980).

**SIMULATION.** In contrast to the previous negative result, we develop a template, SIMULATION, to adapt stochastic algorithms to the random-order model. It partitions the

---

[2]For a formal proof, see Appendix B.

[3]Note, here statistical (offline) learning is equivalent to stochastic online learning, due to the standard online-to-offline reduction.

time horizon into $\log T$ geometrically increasing windows. At the start of each window, past data is used to *simulate* an i.i.d. distribution matching the expected values of the random-order instance, allowing the algorithm to *train* in an i.i.d. setting without incurring real loss. The action frequencies during training are then used on the actual instance within the current block. The analysis leverages that sampling without replacement, even if statistically distinct from i.i.d. sampling, concentrates around the same mean.

### 1.3. Further Related Work

**Random-order Online Convex Optimization.** Random-order inputs in online convex optimization has been studied in Garber et al. (2020) and Sherman et al. (2021). Their main result is that such random-order instances allow for logarithmic regret *even if the single losses are not strongly convex*, as long as the *cumulative* loss is strongly convex.

**Online Combinatorial Problems in Random Order.** Random-order inputs have also been studied in the context of online combinatorial optimization in Dong & Yoshida (2023). They consider approximate follow-the-leader algorithms and prove that when the offline optimization algorithm has low average sensitivity (i.e., the average impact of single points on the output is small, a notion introduced in Varma & Yoshida (2023)), then the offline-to-online reduction carries over also in the random-order model.

**Online Learning with Long-Term Constraints** Online learning under time-varying constraints was first studied by Mannor et al. (2009), who showed that achieving both sublinear regret and constraint violations is impossible when costs and rewards are linear but adversarially generated. Therefore, a significant gap exists between the stochastic setting, where $\widetilde{O}(\sqrt{T})$ regret and constraint violation are achievable (Yu et al., 2017; Badanidiyuru et al., 2013; Agrawal & Devanur, 2014), and the adversarial setting, where guarantees typically take the form of no-$\alpha$-regret bounds, with $\alpha \in (0, 1)$ representing the competitive ratio (Immorlica et al., 2022; Castiglioni et al., 2022a;b; Kesselheim & Singla, 2020; Bernasconi et al., 2024). Some studies achieve improved results under adversarially generated constraints, but at the cost of using a weaker baseline, such as static regret (Sun et al., 2017). Another line of work (Balseiro et al., 2023; Fikioris & Tardos, 2023) bridges the gap between stochastic and adversarial settings by making assumptions about the environment's evolution over time.

## 2. Model

A learner repeatedly chooses one of $k$ actions over a time horizon $T$. At each time step $t$, the loss associated to action $a$ is denoted by $\ell_t(a) \in [0, 1]$. The loss vectors in the random-order model are generated by an adversary and are presented to the learner in random order. The adversary generates a (multi-)set of $T$ loss vectors $h_t \in [0, 1]^k$, then they are presented to the learner uniformly at random, i.e., a permutation $\pi$ is drawn and the loss at time $t$ is $h_{\pi(t)} = \ell_t$. We adopt the notation $[n]$ to denote the set $\{1, 2, \ldots, n\}$, and we denote by $\Delta_n$ the $n$-dimensional probability simplex. The model and results for the online classification are deferred to the Supplementary Materials (Appendix H).

### 2.1. Prediction with Delayed Feedback

In Prediction with Delayed Feedback, the loss vectors are revealed to the learner after a known and fixed delay $d$. More precisely, for any $t$, the learner only observes the loss vector $\ell_t$ after $d$ time steps. We have the following definition of (expected) regret of algorithm $\mathcal{A}$ choosing actions $a_t$ against the random-order input $\mathcal{S}$ of losses $\ell_1, \ldots, \ell_T$:

$$R_T(\mathcal{A}, \mathcal{S}) = \mathbb{E}\left[\sum_{t=1}^{T} \ell_t(a_t) - \min_{a \in [k]} \sum_{t=1}^{T} \ell_t(a)\right]. \quad (1)$$

We denote with regret of the algorithm $\mathcal{A}$ its worst-case regret, i.e., $\sup_{\mathcal{S}} R_T(\mathcal{A}, \mathcal{S})$. Note, the benchmark — i.e., the performance of the best fixed action in hindsight — is not influenced by the realization of the random permutation (in fact, we can equivalently write the benchmark as $\min_a \sum_{t=1}^{T} h_t(a)$). There is a plethora of delayed feedback models studied in the literature; for the sake of simplicity, here we adopt arguably the simplest one with a fixed and known delay $d$, as in, e.g., Masoudian et al. (2022). With minimal effort, our results also apply to other models, e.g., random delay or unknown delay.

### 2.2. Online Learning with Constraints

We consider a setting similar to Balseiro et al. (2023) in which the learner has $m$ resource-consumption constraints. At each $t$, the learner plays $x_t \in \Delta_k$ and subsequently observes a reward vector $r_t \in [0, 1]^k$ and $m$ cost vectors $(c_{1,t}, \ldots, c_{m,t}) \in [0, 1]^{k \times m}$, one for each available resource.[4] The objective of the learner is to maximize the cumulative rewards $\sum_{t=1}^{T} r_t^\top x_t$ while guaranteeing that, for each $j \in [m]$, $\sum_{t=1}^{T} c_{j,t}^\top x_t \leq B$, where $B$ is the available budget for each resource.[5] The per-iteration budget is defined as $\rho = B/T$. We assume that there is a known action $\varnothing$ such that $c_{j,t}(\varnothing) = 0$ for all $j$ and $t$ (this can be thought as a "skipping turn" action, usually employed in the Bandits with Knapsack literature, see e.g., Badanidiyuru et al. (2013); Immorlica et al. (2022)).

---

[4]In the literature on Online Learning with Constraints and Bandits with Knapsacks, it is customary to use rewards instead of losses, and we follow the same notation here.

[5]Assuming that all budgets are the same comes without loss of generality. See discussion in Immorlica et al. (2022).

In the random-order input model, an instance $\mathcal{S}$ is represented by a multi-set of $T$ tuples $(r, c_1, \ldots, c_m)$ presented to the learner in uniform random order. We denote with $\bar{r}$ the average reward of the tuples in $\mathcal{S}$, and, for each resource $j$, we let $\bar{c}_j$ be its average consumption. The natural benchmark for the problem is given by the solution to the following LP (see, e.g., Immorlica et al. (2022)):

$$\mathrm{OPT}_{\mathrm{LP}}(\bar{r}, \bar{c}_1, \ldots, \bar{c}_m) = \begin{cases} \max_{x \in \Delta_k} \langle \bar{r}, x \rangle & \text{s.t.} \\ \langle \bar{c}_j, x \rangle \leq \rho & \forall j \in [m] \end{cases}.$$

With $\mathrm{OPT}_{\mathrm{LP}}$, we denote the solution to the above LP instantiated with the average rewards and costs computed from $\mathcal{S}$. Then, we define the regret in this setting as

$$R_T(\mathcal{A}, \mathcal{S}) = T \cdot \mathrm{OPT}_{\mathrm{LP}} - \sum_{t=1}^{\tau} r_t^\top x_t,$$

where $\tau \in [T]$ is the stopping time of the algorithm (i.e., the time in which the first of the $m$ resources is fully depleted). Once $\tau$ is reached, the algorithm plays $\varnothing$ for the remainder of the time horizon. Therefore, our algorithm enforces resource consumption constraints strictly, similar to what happens in bandits with knapsacks (Immorlica et al., 2022). This is a stronger requirement than ensuring sublinear constraint violations, which is the typical approach in online learning with long-term constraints (Mannor et al., 2009; Yu et al., 2017; Castiglioni et al., 2022b).

### 2.3. Bandits with Switching Costs

In the bandit with switching costs problem, at the end of each time $t$, the algorithm *only* observes its loss $\ell_t(a_t)$ and suffers an additional unitary loss every time it changes action. For instance, this is the same model studied in Cesa-Bianchi et al. (2013); Dekel et al. (2014). The goal is to devise a learning strategy that minimizes the suffered loss or, equivalently, minimizes the regret with respect to the best-fixed action in hindsight. We then have the following definition of (expected regret) of algorithm $\mathcal{A}$ choosing actions $a_t$[6] against the random-order input $\mathcal{S}$ of losses $\ell_1, \ldots, \ell_T$:

$$R_T(\mathcal{A}, \mathcal{S}) = \mathbb{E}\left[\sum_{t=1}^{T} \ell_t(a_t) + \mathbb{I}_{\{a_t \neq a_{t+1}\}} - \min_{a \in [k]} \sum_{t=1}^{T} \ell_t(a)\right].$$

We denote with regret of the algorithm $\mathcal{A}$ its worst-case regret, i.e., $\sup_{\mathcal{S}} R_T(\mathcal{A}, \mathcal{S})$. Note, the benchmark is not influenced by the switching cost or by the random permutation (in fact, we can equivalently write the benchmark as $\min_{a \in [k]} \sum_{t=1}^{T} h_t(a)$).

## 3. The Birthday Paradox

In this section, we analyze a learning algorithm that exhibits the no-regret property against any stochastic input but fails

against a random-order one. We consider the basic Prediction with Experts problem, where the learner immediately observes the loss vector $\ell_t$ upon playing $a_t$. The regret in the stochastic setting is defined with respect to the expected performance of the best fixed arm, while in the random-order model it is defined as in Equation (1) (see also Appendix B for a comparison on the regret definitions).

Consider the following simple BIRTHDAY-TEST algorithm: it repeatedly plays the first action until a certain stopping time $\tau$ is realized, after which it starts to run some no-regret algorithm such as FOLLOW-THE-LEADER (Hannan, 1957) from scratch.[7] The stopping time $\tau$ is defined as the first time step when one of two things happens: either $\ell_t(1) \notin \{i/T, \text{ for } i = 1, 2, \ldots, T\}$, or $\ell_s(1) = \ell_t(1)$ for some $s \neq t$. Intuitively, up to time $\tau$, the algorithm's behavior while playing action 1 is essentially a test to determine whether the losses for this action are drawn uniformly at random from the set $\{i/T \mid i = 1, 2, \ldots, T\}$.

If the underlying distribution is i.i.d., then $\tau$ realizes pretty soon, while it never does under the random-order model. This result is based on folklore calculations related to the birthday paradox, but we report a self-contained proof for completeness in Appendix D.

**Lemma 3.1.** *For any i.i.d. input and $T$ sufficiently large, it holds that $\mathbb{E}[\tau] \leq 2\sqrt{T}$.*

Consider now the performance of BIRTHDAY-TEST against any i.i.d. input: it suffers at most constant instantaneous regret up to the stopping time $\tau$ (for an overall expected regret of $4\sqrt{T}$, as proved in Lemma 3.1) and then run FOLLOW-THE-LEADER, for an overall regret rate of $O(\sqrt{T \log T})$.

It turns out that there exists a random-order instance that fools BIRTHDAY-TEST. The instance has only two actions: action 1, whose losses are given by a permutation of the set $\{i/T, \text{ for } i = 1, 2, \ldots, T\}$, and action 2, which always yields 0 loss. BIRTHDAY-TEST on this instance always plays the first action, accumulating regret at each time step (as $\tau$ will never realize by definition), for a cumulative regret of $\Omega(T)$. All in all, we have proved the following result.

**Theorem 3.2.** *Consider the Prediction with Experts problem. BIRTHDAY-TEST exhibits $O(\sqrt{T \log T})$ regret against any stochastic i.i.d. instance, but there exists a random-order instance against which it suffers $\Omega(T)$ regret.*

## 4. A General Template: SIMULATION

In this Section, we present a general reduction template: SIMULATION. We then show how this general idea can be applied to prediction with delayed feedback, in online learning with constraints, and bandits with switching costs.

---

[6]For simplicity, we define $a_{T+1} = a_T$.

[7]We only need the algorithm to be no-regret on stochastic i.i.d. inputs.

**Algorithm 1** *

  **Input:** stochastic algorithm $\mathcal{A}$
  **for** $i = 0, 1, 2, \ldots, \log T$ **do**
    **(i) iid-ify past data**
      Construct a distribution on past data $\mathcal{D}_i$
    **(ii) Train $\mathcal{A}$ on a simulated past**
      Run algorithm $\mathcal{A}$ over $2^i$ i.i.d. samples of $\mathcal{D}_i$
    **(iii) Test over new data**
      Let $n_i(a)$ be the times that $\mathcal{A}$ plays action $a$ on $\mathcal{D}_i$
      **for** $t = 2^i + 1, 2^i + 2, \ldots, 2^{i+1}$ **do**
        Play action $a_t \sim \left( {}^{n_i(1)}/_{2^i}, \ldots, {}^{n_i(k)}/_{2^i} \right)$
  **end for** The SIMULATION Template

**SIMULATION.** The core idea of our approach (see the pseudocode for details) is to divide the time horizon into blocks of geometrically increasing length, performing three key operations within each block. At the beginning of the $i$-th block, which has a length of roughly $2^i$, we look at the feedback received during the previous time steps and construct a distribution $\mathcal{D}_i$ (for instance, by uniform sampling on the previous $2^i$ samples). Next, we "train" our stochastic algorithm $\mathcal{A}$ on $\mathcal{D}_i$ (without incurring real losses). Finally, we use $\mathcal{A}$'s observed performance over $\mathcal{D}_i$ to play against the actual losses in the time block. We observe that SIMULATION is not a black-box reduction but a general "recipe" for adapting stochastic algorithms to the random-order setting.

To build a high-level intuition on the effectiveness of SIMULATION, we can look at its performance when the underlying stochastic algorithm is BIRTHDAY-TEST. By fictitiously letting play SIMULATION against distribution $\mathcal{D}_i$ we trigger the stopping time $\tau$ so that BIRTHDAY-TEST starts playing as FOLLOW-THE-LEADER on the testing part.

### 4.1. Predictions with Delayed Feedback

In the prediction with delayed feedback model, the learner has access to the loss vector after $d$ time steps. To specialize the SIMULATION template to this model, we need to account for the delayed feedback by adding a buffer of $d$ time steps between the blocks, allowing enough time to receive the feedback corresponding to the previous block (we refer to the pseudocode for further details).

At the beginning of the generic time block $i$, the distribution $\mathcal{D}_i$ that is used to simulate past data is simply the uniform distribution over all the loss vectors observed in the blocks before the $i^{th}$ which are contained in the multiset $O$ (the losses observed in $i \cdot d$ time steps corresponding to the previous buffers are discarded). To simplify the notation, we denote with $T_i$ the time steps in the $i^{th}$ time block, i.e., $T_i = \{t_i, \ldots, t_i + 2^i - 1\}$, where $t_i = 2^i + i \cdot d$.

Fix any sequence of losses and any block $i$. At the beginning of $T_i$, the multiset $O$ contains all the losses observed in the

**Algorithm 2** *

  **Input:** stochastic algorithm $\mathcal{A}$
  **Environment**: $K$ actions, time horizon $T$
  **Model**: full feedback model with delay $d$.
  Play any action at time 1
  Initialize multiset $O = \{\ell_1\}$
  **for** $i = 0, 1, 2, \ldots, \lceil \log T \rceil$ **do**
    Let $\mathcal{D}_i$ be the uniform distribution on $O$ {iid-ify past}
    Run $\mathcal{A}$ over $2^i$ i.i.d. samples of $\mathcal{D}_i$   {Training}
    Let $n_i(a)$ be the number of times that $\mathcal{A}$ plays $a$
    $t_i \leftarrow i \cdot d + 2^i$     {Beginning of $i^{th}$ block}
    **for** $t = t_i, t_i + 2, \ldots, t_i + 2^i - 1$ **do**
      Play action $a_t \sim \left( {}^{n_i(1)}/_{2^i}, \ldots, {}^{n_i(k)}/_{2^i} \right)$   {Test}
      Add loss $\ell_t$ to $O$ when revealed
      **if** $t = T$, **then** terminate
    **end for**
    Play arbitrarily for the next $d$ time steps
  **end for** SIMULATION for delayed feedback

previous time blocks, which are $2^i$. Therefore, sampling according to $\mathcal{D}_i$ yields an unbiased estimator of the average loss in the previous blocks. This is formalized in the following Lemma.

**Lemma 4.1.** *Fix any sequence of losses and time block $i$. Then we have:*

$$\mathbb{E}_{\ell \sim \mathcal{D}_i} [\ell(a)] = \frac{1}{2^i} \sum_{i' < i} \sum_{t \in T_{i'}} \ell_t(a) \quad \forall a \in [k].$$

In the analysis of SIMULATION for prediction with delayed feedback, we employ a generic stochastic routine $\mathcal{A}$ which is guaranteed an i.i.d. regret bound of $R_{T'}^{\text{iid}}(\mathcal{A})$ against any stochastic input of length $T'$.

**Theorem 4.2.** *Consider the problem of online prediction with delayed feedback in the random-order model, and let $d$ be the delay parameter. Running SIMULATION for delayed feedback with stochastic routine $\mathcal{A}$ (SIM-$\mathcal{A}$) yields the following regret bound:*

$$R_T(\text{SIM-}\mathcal{A}) \leq 5\sqrt{T \log T} + d \log T + \sum_{i=0}^{\log T} R_{2^i}^{\text{iid}}(\mathcal{A}).$$

*Proof.* Fix any input sequence $\mathcal{S} = \{h_1, \ldots, h_T\}$, any realization of the permutation $\{\ell_1, \ldots, \ell_T\}$, and denote with $a^\star$ the corresponding best action ($a^\star$ only depends on $\mathcal{S}$, not on the specific permutation). For any block $i$, the stochastic algorithm $\mathcal{A}$ is trained over $2^i$ i.i.d. samples from $\mathcal{D}_i$, and the empirical frequencies ${}^{n_i(a)}/_{2^i}$ are then used to play in $T_i$. For any action $i$, denote with $\overline{\Delta}_i(a)$ the gap between the loss of action $a$ and that of action $a^\star$ according to $\mathcal{D}_i$:

$$\overline{\Delta}_i(a) = \mathbb{E}_{\ell \sim \mathcal{D}_i} [\ell(a) - \ell(a^\star)].$$

From the guarantees on the regret bound of $\mathcal{A}$, we get

$$\sum_a \mathbb{E}[n_i(a)] \overline{\Delta}_i(a) \leq R_{2^i}^{\text{iid}}(\mathcal{A}), \tag{2}$$

where the expectation is with respect to the randomness in the i.i.d. sampling from $\mathcal{D}_i$.

The gaps $\overline{\Delta}_i(a)$ induced by the distribution $\mathcal{D}_i$ are related to the empirical gaps $\Delta_i^p(a)$ experienced in the previous time blocks by Lemma 4.1. In particular, for any $\Delta_i^p(a)$ (the superscript stands for "past") we have:

$$
\begin{aligned}
\Delta_i^p(a) &= \frac{1}{2^i} \sum_{i' < i} \sum_{t \in T_{i'}} (\ell_t(a) - \ell_t(a^\star)) && \text{(By definition)} \\
&= \mathbb{E}_{\ell \sim \mathcal{D}_i} [\ell(a) - \ell(a^\star)] && \text{(By Lemma 4.1)} \\
&= \overline{\Delta}_i(a). && \text{(By definition of } \overline{\Delta}_i(a))
\end{aligned}
$$

Plugging in the above equality into the stochastic regret guarantees as in Equation (2), we get:

$$
\sum_a \mathbb{E}[n_i(a)] \, \Delta_i^p(a) \leq R_{2^i}^{\mathrm{iid}}(\mathcal{A}). \tag{3}
$$

While the above inequality characterizes the performance of SIMULATION if run on "past" losses, we need to relate it to the "future" or actual ones, i.e., the ones appearing in time block $i$. We denote with $\Delta_i(a)$ the gaps in the current time block. Let's now look at the actual regret suffered by the algorithm during $T_i$:

$$
\begin{aligned}
&\sum_{t \in T_i} (\mathbb{E}[\ell_t(a_t)] - \ell_t(a^\star)) \\
&= \frac{1}{2^i} \sum_a \mathbb{E}[n_i(a)] \sum_{t \in T_i} (\ell_t(a) - \ell_t(a^\star)) && \text{(By design)} \\
&= \sum_a \mathbb{E}[n_i(a)] \, \Delta_i(a) && \text{(By definition of } \Delta_i(a)) \\
&= \sum_a \mathbb{E}[n_i(a)] \, \Delta_i^p(a) + \sum_a \mathbb{E}[n_i(a)] \, (\Delta_i(a) - \Delta_i^p(a)) \\
&\leq R_{2^i}^{\mathrm{iid}}(\mathcal{A}) + 2^i \cdot \max_a |\Delta_i(a) - \Delta_i^p(a)|, \tag{4}
\end{aligned}
$$

where the last inequality follows because of Equation (3) and the fact that $\sum_{a \in [k]} n_i(a) = 2^i$. So far, our argument works for any input model, as the sequence of losses was arbitrary; now it's time to exploit the fact that random-order sequences exhibit concentration. In particular, we want to argue that, for any block $i$ and action $a$, both the past and future empirical gaps $\Delta_i^p(a)$ and $\Delta_i(a)$ are close to the *actual* gap $\Delta(a)$, where

$$
\Delta(a) = \frac{1}{T} \sum_{i \in [T]} (h_i(a) - h_i(a^\star)) = \frac{1}{T} \sum_{t \in [T]} (\ell_t(a) - \ell_t(a^\star)).
$$

We introduce the precision $\epsilon_i = 2\sqrt{\log T / 2^i}$ and the corresponding clean event $\mathcal{E}_i$:

$$
\mathcal{E}_i = \{\max\{|\Delta_i(a) - \Delta(a)|, |\Delta_i^p(a) - \Delta(a)| \leq \epsilon_i, \forall a\}.
$$

We can show that such event is realized with high probability (see Appendix E for omitted proofs).

**Claim 4.3.** *For any block $i$, the corresponding clean event is realized with probability at least $1 - 1/T$.*

Then, conditioning on the clean event $\mathcal{E}_i$, we can improve the bound in Equation (4):

$$
\begin{aligned}
&\sum_{t \in T_i} (\mathbb{E}[\ell_t(a_t)] - \ell_t(a^\star)) - R_{2^i}^{\mathrm{iid}}(\mathcal{A}) \\
&\qquad \leq 2^i \cdot \max_a |\Delta_i(a) - \Delta_i^p(a)| \\
&\qquad \leq 2^{i+1} \max_a \{|\Delta(a) - \Delta_i^p(a)|, |\Delta(a) - \Delta_i(a)|\} \\
&\qquad \leq 4\sqrt{\log T \cdot 2^i}.
\end{aligned}
$$

By removing the conditioning w.r.t. the clean event (and using the probability bound as in Claim 4.3 and the fact that the regret suffered under the bad event is at most $2^i$), we get:

$$
\sum_{t \in T_i} (\mathbb{E}[\ell_t(a_t) - \ell_t(a^\star)]) \leq R_{2^i}^{\mathrm{iid}}(\mathcal{A}) + 5\sqrt{\log T \cdot 2^i}.
$$

We are ready to bound the overall regret: for any time block, we have the above inequality, while for the $\log T$ buffers, we suffer an overall regret of at most $d \log T$. All in all, we have that the regret $R_T$ of SIMULATION for delayed feedback with routine $\mathcal{A}$ is at most:

$$
\begin{aligned}
R_T(\text{SIM-}\mathcal{A}) &\leq \sum_{i=0}^{\log T} \left( R_{2^i}^{\mathrm{iid}}(\mathcal{A}) + 5\sqrt{\log T \cdot 2^i} + d \right) \\
&\leq 5\sqrt{T \log T} + d \log T + \sum_{i=0}^{\log T} R_{2^i}^{\mathrm{iid}}(\mathcal{A}).
\end{aligned}
$$

This concludes the proof. $\qquad \square$

In particular, if we instantiate algorithm $\mathcal{A}$ to be FOLLOW-THE-LEADER, we get the following guarantees.

**Corollary 4.4.** *There exists an algorithm for online prediction with delayed feedback in the random-order model that enjoys an $O(\sqrt{T \log T} + d \log T)$ regret bound.*

### 4.2. Online Learning with Constraints

In the Online Learning with Constraints model, the learner has to play against an environment in which each action $a \in [k]$ has an associated reward and a vector of costs. We analyze the problem within the random-order model and show that, in this setting, it is possible to achieve regret guarantees comparable to those in the stochastic case, despite impossibility results for the adversarial setting (Mannor et al., 2009).

Here, we show how we can adapt SIMULATION in the presence of long-term constraints. The main difficulty in adapting SIMULATION in this setting is the fact that the costs

**Algorithm 3** *
> **Input:** stochastic algorithm $\mathcal{A}$, budget $B$, parameter $\delta$
> **Environment**: $K$ actions, time horizon $T$
> Play any action at time 1
> Initialize $O = \{r_1, c_1^1, \ldots, c_1^m\}$
> **for** $i = 0, 1, 2, \ldots, \log T$ **do**
> > Let $\mathcal{D}_i$ be the uniform distribution over $O$ {iid-ify past}
> > Run algorithm $\mathcal{A}$ over $2^i$ i.i.d. samples of $\mathcal{D}_i$ with budget $\rho - \widetilde{O}(2^{-i/2})$ and $\delta' = \delta/3\log T$ {Training}
> > Let $n_i(a)$ be the number of times that $\mathcal{A}$ plays action $a$. Compute $x_i(a)$ as $n_i(a)/2^i$ for each action $a$
> > **for** $t = 2^i + 1, 2^i + 2, \ldots, 2^{i+1}$ {Play against the actual sequence} **do**
> > > Play $x_i$ and subsequently add $(r_t, c_t^1, \ldots, c_t^m)$ to $O$
> > **end for**
> **end for** SIMULATION for Online Learning with Constraints

impose constraints on the entire time horizon, while SIMU-LATION naturally works on each block independently.

To run SIMULATION in this setting, we need access to an algorithm $\mathcal{A}$ for stochastic online learning with constraints (see, e.g., Castiglioni et al., 2022a; Bernasconi et al., 2024). In particular, in a stochastic environment with mean reward $\tilde{r}$ and mean costs $(\tilde{c}_1, \ldots, \tilde{c}_m)$, $\mathcal{A}$ guarantees a regret upper bound, with probability at least $1 - \delta$, of

$$T \cdot \mathrm{OPT}_{\mathrm{LP}}(\tilde{r}, \tilde{c}_1, \ldots, \tilde{c}_m) - \sum_{t=1}^{\tau} \tilde{r}^\top x_t \leq R_T^{\mathrm{iid}, \delta}(\mathcal{A}, \rho),$$

where $\tau$ denotes the first time in which a resource $j$ is fully depleted, that is $\sum_{t=1}^{\tau} c_{j,t}^\top x_t > \rho T$.[8]

In each block $i$ we train $\mathcal{A}$ on i.i.d. samples $(r_t^{\mathrm{iid},i}, c_{t,1}^{\mathrm{iid},i}, \ldots, c_{t,m}^{\mathrm{iid},i})_t \sim \mathcal{D}_i$.[9] Then, we take the empirical frequency of play $x_i$ obtained in the training step and use it against the actual environment in the time block $2^i + 1, \ldots, 2^{i+1}$.

Crucially, during each training phase, we instantiate $\mathcal{A}$ with a slightly reduced per-iteration budget of $\rho - \widetilde{O}(\sqrt{2^{-i}})$. This allows us to control concentration terms in the analysis. Then, one of the main challenges is proving that this budget rescaling does not significantly affect the guarantees of the algorithm $\mathcal{A}$ during the simulation phase.

**Theorem 4.5.** *Consider the problem of online learning with long-term constraints in the random-order model. Running* SIMULATION *with stochastic routine $\mathcal{A}$ (SIM-$\mathcal{A}$) yields the*

---

*following regret bound with probability at least $1 - \delta$,*

$$R_T(\mathrm{SIM}\text{-}\mathcal{A}) \leq O\left(\frac{\log(\frac{mkT}{\delta})}{\rho^2} + \frac{\sqrt{T\log\left(\frac{mk}{\delta}\right)}}{\rho}\right)$$
$$+ \sum_{i=1}^{\log T} R_{2^i}^{\mathrm{iid},\delta'}(\mathcal{A}, \rho/2).$$

*where $\delta' \in \widetilde{O}(\delta)$,*

*Proof.* Fix any sequence $\mathcal{S} \in [0,1]^{T \times k \times m + 1}$ and any permutation $\{(r_t, c_{t,1}, \ldots, c_{t,m})\}_{t \in [T]}$. Let $x^\star$ be the solution to $\mathrm{OPT}_{\mathrm{LP}}$. For each time block $i$ we are going to define the following quantities: $r_i = \frac{1}{2^i}\sum_{t=2^i+1}^{2^{i+1}} r_t$ is the empirical mean of the rewards on the "next" block, $r_i^p = \frac{1}{2^i}\sum_{t=1}^{2^i} r_t$ is the empirical mean of rewards in the "past" block, and $\bar{r}_i = \mathbb{E}_{\mathcal{D}_i}[r]$ is the expected reward w.r.t. $\mathcal{D}_i$. Similarly we define $c_{i,j}, c_{i,j}^p$ and $\bar{c}_{i,j}$ for each resource $j \in [m]$. Clearly, $\bar{r}_i = r_i^p$ and $\bar{c}_{i,j} = c_{i,j}^p$.

We can define the benchmark $\mathrm{OPT}_{\mathrm{LP}}^i$ on the past block as

$$\max_{x \in \Delta_k} x^\top r_i^p \quad \text{s.t.} \quad \max_{j \in [m]} x^\top c_{i,j}^p \leq \rho - 2 \cdot \epsilon_i = \rho_i,$$

where $\epsilon_i = \sqrt{6 \cdot 2^{-i}\log(mk\log(T)/\delta)} \in \widetilde{O}(\sqrt{2^{-i}})$. We define the event $\mathcal{E}_i = \{\|\bar{r}_i - \bar{r}\|_\infty \leq \epsilon_i, \|r_i - \bar{r}\|_\infty \leq \epsilon_i, \|\bar{c}_{i,j} - \bar{c}_j\|_\infty \leq \epsilon_i, \|c_{i,j} - \bar{c}_j\|_\infty \leq \epsilon_i \ \forall j \in [m]\}$. From now on, we condition on the event $\mathcal{E}_i$, which we show to hold with high probability with respect to the permutation.

The regret of SIMULATION can be written as

$$R_T(\mathrm{SIM}\text{-}\mathcal{A}, \mathcal{S}) = \sum_{i=1}^{\log T}\left(2^i \cdot \bar{r}^\top x^\star - \sum_{t=2^i+1}^{2^{i+1}} r_t^\top x_t\right)$$
$$= \sum_{i=1}^{\log T} 2^i \cdot \left(\bar{r}^\top x^\star - r_i^\top x_i\right), \quad (5)$$

where the second equality holds thanks to the fact that we are playing the fixed learned distribution $x_i$ in the $i$-th block, which is the one learned while playing against $\mathcal{D}_i$.

We only consider blocks with index $i > i^\star$ where we set $i^\star = \Omega(\log(\log(mk\log(T)/\delta)/\rho^2))$, otherwise we would have $\rho_i < 0$ which is the first block for which $\rho_i = \rho - 2\epsilon_i \geq \rho/2$. For each block $i < i^\star$ we suffer linear regret (since we are forced to play the null action $\varnothing$), and this contributes to a $\sum_{i=1}^{i^\star} 2^i \in \widetilde{O}(1/\rho^2)$ additional regret term.

The no-regret properties of $\mathcal{A}$ guarantee that, with probability at least $1 - \delta'$:

$$2^i \cdot \mathrm{OPT}_{\mathrm{LP}}^i - \sum_{t=1}^{2^i} x_t^\top \bar{r}_i = 2^i \cdot \left(x_i^{\star,\top} r_i^p - x_i^\top \bar{r}_i\right)$$
$$\leq R_{2^i}^{\mathrm{iid},\delta'}(\mathcal{A}, \rho - 2 \cdot \epsilon_i),$$

where $x_i^\star$ is the solution to $\text{OPT}_{\text{LP}}^i$ and

$$\sum_{s=1}^t x_s^\top c_{s,j}^{\text{iid}} \le (\rho - \epsilon_i) \cdot 2^i \tag{6}$$

which holds for all $t \in [2^i]$ and for all $j \in [m]$.

Consider one term of the regret decomposition of Equation (5):

$$\bar{r}^\top x^\star - r_i^\top x_i = (\bar{r}^\top x^\star - r_i^{p,\top} x_i^\star) + (r_i^{p,\top} x_i^\star - r_i^\top x_i)$$

$$\le (\bar{r}^\top x^\star - r_i^{p,\top} x_i^\star) + \frac{R_{2^i}^{\text{iid},\delta}(\mathcal{A}, \rho_i)}{2^i} + (\bar{r}_i - r_i)^\top x_i \tag{7}$$

where the last inequality holds by the regret properties of $\mathcal{A}$. Now, we have to analyze the first and last terms of the inequality above. The last term is easy to bound. Indeed, conditioning on $\mathcal{E}_i$ it holds that

$$(\bar{r}_i - r_i)^\top x_i \le \|\bar{r}_i - \bar{r}\|_\infty + \|\bar{r} - r_i\|_\infty \le 2\epsilon_i.$$

Bounding the first term is trickier as it is equivalent to bounding the difference between the values of two LPs. Indeed, it can be rewritten as $\text{OPT}_{\text{LP}} - \text{OPT}_{\text{LP}}^i$. The following claim (whose proof can be found in Appendix F) shows that this difference is not too large.

**Claim 4.6.** *For all $i > i^\star$, under the event $\mathcal{E}_i$, we have that*

$$\text{OPT}_{\text{LP}} - \text{OPT}_{\text{LP}}^i \le \epsilon_i \left(1 + \frac{3}{\rho}\right).$$

Plugging everything back into Equation (7) we get that, conditioning on the events $\mathcal{E}_i$,

$$R_T \le \sum_{i=1}^{i^\star} 2^i + \sum_{i=i^\star+1}^{\log T} \left(\epsilon_i \left(1 + \frac{3}{\rho}\right) + 2\epsilon_i + R_{2^i}^{\text{iid},\delta'}(\mathcal{A}, \rho_i)\right)$$

$$\le O\left(\frac{\log(\frac{mkT}{\delta})}{\rho^2}\right) + \frac{6}{\rho} \sum_{i=1}^{\log T} \epsilon_i + \sum_{i=i^\star}^{\log T} R_{2^i}^{\text{iid},\delta'}(\mathcal{A}, \rho_i)$$

$$\le O\left(\frac{\log(\frac{mkT}{\delta})}{\rho^2} + \frac{\sqrt{T \log\left(\frac{mk}{\delta}\right)}}{\rho}\right) + \sum_{i=1}^{\log T} R_{2^i}^{\text{iid},\delta'}\left(\mathcal{A}, \frac{\rho}{2}\right),$$

where we used that $\rho \mapsto R_t^{\text{iid},\delta}(\mathcal{A}, \rho)$ has to be monotone decreasing and positive. Moreover, note that the probability measure is the one induced by the random permutation and does not depend on the algorithm.

Next, we analyze the violations, which for each $t$ and each

resource $j \in [m]$ are given by:

$$\sum_{s=1}^t x_s^\top c_{s,j} \le \sum_{i=1}^{\lceil \log t \rceil} \sum_{s=2^i+1}^{2^{i+1}} x_i^\top c_{s,j} = \sum_{i=1}^{\lceil \log t \rceil} 2^i \cdot c_{i,j}^\top x_i$$

$$= \sum_{i=1}^{\lceil \log t \rceil} 2^i \cdot \bar{c}_{i,j}^\top x_i + \sum_{i=1}^{\lceil \log t \rceil} 2^i \cdot (c_{i,j} - \bar{c}_{i,j})^\top x_i$$

$$\le \sum_{i=1}^{\lceil \log t \rceil} 2^i \cdot \bar{c}_{i,j}^\top x_i + \sum_{i=1}^{\lceil \log t \rceil} 2^i \epsilon_i, \tag{8}$$

where the last inequality holds by conditioning on all $\mathcal{E}_i$. To analyze the first term, we can consider the following sequence of random variables $Z_t^{i,j} = \sum_{s=1}^t (c_{s,j}^{\text{iid}} - \bar{c}_{i,j})^\top x_s$. This is a martingale sequence (since $\mathbb{E}_{\mathcal{D}_i}[c_{s,j}^{\text{iid}}] = \bar{c}_{i,j}$ and has differences bounded by 1) with $Z_0^{i,j} = 0$ and thus, by Azuma-Hoeffding, we have that with probability at least $1 - \frac{\delta}{3m \log T}$ with respect to the randomness of $\mathcal{D}_i$:

$$|Z_{2^i}^{i,j}| = \left|\sum_{s=1}^{2^i}(c_{s,j}^{\text{iid}} - \bar{c}_{i,j})^\top x_s\right| \le 2^i \epsilon_i,$$

and thus for all $i \in [\log T]$ and $j \in [m]$:

$$2^i \cdot \bar{c}_{i,j}^\top x_i \le 2^i \epsilon_i + \sum_{s=1}^{2^i} c_{s,j}^{\text{iid},\top} x_s \le 2^i \epsilon_i + 2^i(\rho - 2 \cdot \epsilon_i)$$

with probability at least $1 - \delta/3$ (by union bound on all $i \in [\log T]$ and $j \in [m]$), and the second inequality holds thanks to the properties of $\mathcal{A}$ (Equation (6)). Plugged back into Equation (8) we get that

$$\sum_{s=1}^t x_s^\top c_{s,j} \le \sum_{i=1}^{\lceil \log t \rceil} (2^i \epsilon_i + 2^i \epsilon_i + 2^i(\rho - 2 \cdot \epsilon_i)) \le \rho t.$$

Finally, we prove that all of the previous results hold with probability at least $1 - \delta$. We have three sources of randomness; the first is due to the permutation, which is encoded by the events $\mathcal{E}_i$. The following claim proves that these hold with high probability.[10]

**Claim 4.7.** *The event $\cap_{i=1}^{\log T} \mathcal{E}_i$ holds with probability at least $1 - \delta/3$.*

Second, we have the randomness of the i.i.d. sampling. By Azuma-Hoeffding inequality, we proved that this holds with probability at least $1 - \delta/3$. Finally, the algorithm $\mathcal{A}$ has guarantees with probability $\delta' = 1 - \delta/3 \log(T)$ each time it is run on a block $i$. By a further union bound, we have that these events hold jointly with probability $1 - \delta$. This concludes the proof. $\qquad\square$

---

[10]The proof is similar to the proof of Claim 4.3 by relying on Theorem C.1 with additional union bounds on the constraints.

In particular, we can instantiate $\mathcal{A}$ to be the algorithm of Castiglioni et al. (2022a) that guarantees that $R_T^{\text{iid},\delta}(\mathcal{A}, \rho) \in O(1/\rho\sqrt{T \log(kmT/\delta)})$, and we can prove the following:[11]

**Corollary 4.8.** *There exists an algorithm for online learning with long-term constraints for the random-order model with regret $\widetilde{O}(1/\rho^2 + \sqrt{T}/\rho)$ with high probability.*

Our reduction maintains the same regret guarantees as $\mathcal{A}$, but for an extra $1/\rho^2$ instead of $1/\rho$. However, this term does not depend on $T$, excluding poly-logarithmic factors.

### 4.3. Bandits With Switching Costs

We present an algorithm for the Bandits with Switching Costs problem that suffers $O(\sqrt{T})$ regret in the random-order model, as opposed to the adversarial setting, where the minimax regret rate is $\Theta(T^{2/3})$ (Dekel et al., 2014).

Our algorithm SIMULATION-SUCCESSIVE-ELIMINATION maintains a set of active actions and proceeds in geometrically increasing time blocks. In each time block, it simply plays in a low-switching round-robin way the actions that are still active; then it updates the confidence bounds on the actions played and discards the ones that are proved to be, with high probability, suboptimal. We refer to the pseudocode for more details. Note, SIMULATION-SUCCESSIVE-ELIMINATION is an application of the SIM-ULATION paradigm to this problem, using as stochastic subroutine SUCCESSIVE-ELIMINATION (e.g., Chapter 1 of Slivkins, 2019): running SUCCESSIVE-ELIMINATION on the i.i.d. version of past data quickly converges to playing round-robin on the active actions. For simplicity, we analyze this simpler version of the algorithm, already specialized for the routine SUCCESSIVE-ELIMINATION. We note that this algorithm is similar to a low-switching algorithm in the literature (Agrawal et al., 1988) that achieves $\sqrt{T}$ regret in the stochastic case. It shares the geometrically increasing blocks and the idea of playing each active arm in contiguous intervals in each block; the main difference is that they use UCB as a routine to choose the next action, while our algorithm uses SUCCESSIVE-ELIMINATION to simplify the analysis for the random-order input model.

We state here the main result concerning SIMULATION-SUCCESSIVE-ELIMINATION. We defer the complete proof to the Supplementary Materials (Appendix G).

**Theorem 4.9.** *There exists an algorithm for bandits with switching costs in the random-order model with regret $O(\sqrt{kT \log^3 T})$.*

*Proof Sketch.* SIMULATION-SUCCESSIVE-ELIMINATION switches actions at most $O(k)$ times during each one of

---

**Algorithm 4** *

   **Environment**: $k$ actions and time horizon $T$
   **Model**: bandit feedback with switching costs
   $\hat{\ell}(a) \leftarrow 0, n(a) \leftarrow 0$, for all actions $a$
   $A \leftarrow [k]$ set of active actions
   $i_0 \leftarrow \lceil \log k \rceil$
   **for** $t = 1, \ldots 2^{i_0}$ **do**
      Play $a_t = t-1$ modulo $k$    {Play each action once}
      $n(a_t) \leftarrow n(a_t) + 1$
      $\hat{\ell}(a_t) \leftarrow \frac{1}{n(a_t)} \sum_{s=1}^{t} \mathbb{I}_{\{a_s = a_t\}} \ell_s(a_s)$
   **end for**
   **for** $i = i_0, \ldots, \log T$ **do**
      Let $\epsilon_i \leftarrow \sqrt{10k \log^3 T \cdot 2^{-i}}$   {Precision for block $i$}
      **for** $a \in A$ **do**
         Play action $a$ action for $2^i/|A|$ rounds
         Let $\hat{\ell}(a) \leftarrow \frac{1}{n(a)} \sum_{s=1}^{t} \mathbb{I}_{\{a_s = a\}} \ell_s(a)$
      **end for**
      Play last active action $a$ up to time $\min\{2^{i+1} - 1, T\}$
      Update accordingly $n(a)$ and $\hat{\ell}(a)$
      Remove from $A$ all actions $a$ such that

$$\hat{\ell}(a) - \epsilon_i > \min_{a' \in A}(\hat{\ell}(a') + \epsilon_i)$$

   **end for** SIMULATION-SUCCESSIVE-ELIMINATION

---

the $O(\log T)$ phases. Therefore, the overall switching cost is always $O(k \log T)$. We move our attention to the regret incurred by playing suboptimal actions. We can argue by concentration for random-order sequences (see Appendix C) that in the generic phase $i$ only actions with overall suboptimality gap $O(\epsilon_{i-1})$ are still active, with high probability. Moreover, once again by concentration, we can argue that these actions' actual (permuted) losses during the generic block $i$ are "close" to their actual suboptimality gap. Summing up for all blocks and plugging in our choice of the precision parameters $\epsilon_i$ yields the desired regret bound. $\square$

## 5. Conclusion and Further Directions

We study online learning in the random-order model, which lies between the adversarial and i.i.d. settings, contributing to the analysis of problems beyond worst-case instances. We show that stochastic algorithms may fail in the random-order model and, to address this, we propose SIMULATION, a general template for designing effective random-order algorithms. We study various relevant online learning problems and prove that random-order and stochastic inputs share the same minimax regret, improving on the adversarial case. While SIMULATION successfully provides black-box reductions for full feedback, the bandit model still requires a more specialized approach. Providing black-box constructions for broader feedback settings, including bandit feedback, is an interesting avenue for future research.

---

[11]The algorithm of Castiglioni et al. (2022a) is, in its turn, instantiated with ONLINE-MIRROR-DESCENT with negative entropy as both the primal and dual algorithm.

## Acknowledgements

The work of MB, AC, FF, SL, and MR was partially funded by the European Union. Views and opinions expressed are however those of the author(s) only and do not necessarily reflect those of the European Union or the European Research Council Executive Agency. Neither the European Union nor the granting authority can be held responsible for them.

AC is partially supported by MUR - PRIN 2022 project 2022R45NBB funded by the NextGenerationEU program and by the ERC grant Project 101165466 – PLA-STEER. FF, SL, MB and MR are also partially supported by the FAIR (Future Artificial Intelligence Research) project PE0000013, funded by the NextGenerationEU program within the PNRR-PE-AI scheme (M4C2, investment 1.3, line on Artificial Intelligence). FF, SL, and MR are supported by the Meta/Sapienza project on "Online Constrained Optimization and Multi-Calibration in Algorithm and Mechanism Design" and by the PNRR MUR project IR0000013-SoBigData.it. SL and MR are also partially supported by the MUR PRIN grant 2022EKNE5K (Learning in Markets and Society).

## Impact Statement

This paper presents work whose goal is to advance the field of Machine Learning. There are many potential societal consequences of our work, none of which we feel must be specifically highlighted here.

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

## A. The De Finetti Theorem

The De Finetti Theorem states that any (infinite) sequence of exchangeable[12] random variables behaves (in distribution) as a mixture of i.i.d. distribution. This result has been initially proven for binary random variables in De Finetti (1929) (a translated version is available on the arXiv Alvarez-Melis & Broderick (2015)), and then generalized to real random variables in De Finetti (1937); Hewitt & Savage (1955). However, this representation theorem does not hold (de Finetti, 1969; Diaconis & Freedman, 1980) for finite sequences. In particular, Diaconis & Freedman (1980) provides tight quantitative bounds on the difference between exchangeable sequences and their "closest" mixture of i.i.d. distributions. We report here a lower bound on such distance.

Let $U_{2n}$ be an urn containing $n$ red and $n$ black balls. Let $H_{2n,2k}$, respectively $M_{2n,2k}$, represent the distribution of the number of red balls out of $2k$ draws made at random without, respectively with, replacement from $U_{2n}$. Furthermore, for any random variable $P$ in $[0,1]$, consider the random experiment that consists of first sampling $p$ according to $P$ and then sampling $k$ i.i.d. Bernoulli with parameter $p$; denote with $\mathbb{P}_{P,k}$ the distribution of the number of successes according to this experiment. We have the following Theorem (Proposition 31 and Theorem 40 of Diaconis & Freedman (1980)):

**Theorem A.1.** *Let $n$ tend to $\infty$, and $k \in o(n)$. Then, for any random variable $P$ in $[0,1]$, the following inequality holds:*

$$||H_{2n,2k} - \mathbb{P}_{P,k}||_{\mathrm{TV}} \geq ||H_{2n,2k} - M_{2n,2k}||_{\mathrm{TV}} = \sqrt{\frac{2}{\pi e} \frac{k}{n}} + O\left(\frac{k}{n}\right).$$

Stated differently, sampling with replacement and without replacement induce different distributions whose total variation distance roughly scales linearly in the sampled fraction of the urn.

## B. The Hierarchy between the Three Models

The random-order model is intermediate between the i.i.d. and adversarial input model. We provide here a formal proof of this fact for the standard online learning setting with $k$ actions and loss minimization.

**Stochastic i.i.d. Input Model.** The input $\mathcal{S}$ is defined by a distribution $\mathcal{D}$ over loss vectors, from which $T$ i.i.d. samples are drawn. The regret suffered by algorithm $\mathcal{A}$ on input $\mathcal{S}$ is defined as follows:

$$R_T^{\mathrm{i.i.d.}}(\mathcal{A}, \mathcal{S}) = \max_{a \in [k]} \mathbb{E}\left[\sum_{t=1}^{T} \ell_t(a_t) - \ell_t(a)\right].$$

The stochastic regret $R_T^{\mathrm{i.i.d.}}(\mathcal{A})$ of an algorithm $\mathcal{A}$ is its worst-case regret against all stochastic inputs.

**Random-Order Model.** The input $\mathcal{S}$ is defined by a set $\{h_1, \ldots, h_T\}$ of $T$ loss vectors, that are presented in random order to the learner, so that the generic $\ell_t$ is $h_{\pi(t)}$ for a random permutation $\pi$. The regret suffered by algorithm $\mathcal{A}$ on input $\mathcal{S}$ is defined as follows:

$$R_T^{\mathrm{RO}}(\mathcal{A}, \mathcal{S}) = \max_{a \in [k]} \mathbb{E}\left[\sum_{t=1}^{T} \ell_t(a_t) - \ell_t(a)\right] = \mathbb{E}\left[\sum_{t=1}^{T} \ell_t(a_t)\right] - \min_{a \in [k]} \sum_{t=1}^{T} h_t(a).$$

The random-order regret $R_T^{\mathrm{RO}}(\mathcal{A})$ of an algorithm $\mathcal{A}$ is its worst-case regret against all random-order inputs.

**(Oblivious) Adversarial Model.** The input $\mathcal{S}$ is defined by a sequence of $T$ loss vectors. The regret suffered by algorithm $\mathcal{A}$ on input $\mathcal{S}$ is defined as follows:

$$R_T^{\mathrm{ADV}}(\mathcal{A}, \mathcal{S}) = \max_{a \in [k]} \mathbb{E}\left[\sum_{t=1}^{T} \ell_t(a_t) - \ell_t(a)\right] = \mathbb{E}\left[\sum_{t=1}^{T} \ell_t(a_t)\right] - \min_{a \in [k]} \sum_{t=1}^{T} \ell_t(a).$$

The adversarial regret $R_T^{\mathrm{ADV}}(\mathcal{A})$ of an algorithm $\mathcal{A}$ is its worst-case regret against all adversarial inputs.

---

[12]A sequence $X_1, X_2, \ldots, X_n$ of random variables is exchangeable if for any finite set of indices $I = (i_1, \ldots, i_\ell)$, and any permutation $\pi$ over $I$ it holds that $(X_{i_1}, \ldots, X_{i_\ell})$ is distributed as $(X_{\pi(i_1)}, \ldots, X_{\pi(i_\ell)})$.

**No-regret.** An algorithm is said to enjoy the no-regret property in a certain input model if it exhibits sublinear regret (in the time horizon $T$) uniformly over all possible inputs in that model.

The informal statement contained in the main body that *i.i.d.* $\preceq$ *random-order* $\preceq$ *adversarial* is formalized in the following theorem.

**Theorem B.1.** *For any learning algorithm $\mathcal{A}$, the following regret hierarchy holds:*

$$R_T^{i.i.d.}(\mathcal{A}) \leq R_T^{RO}(\mathcal{A}) \leq R_T^{ADV}(\mathcal{A}).$$

*Proof.* We start with the first inequality of the statement. Consider any random instance $\mathcal{S}$, and let $X \subseteq [0,1]^k$ be the support of the random variable underlying $\mathcal{S}$,[13] and denote with $\Sigma_X$ the multiset of all the subsets of $T$ elements in $X$ (possibly with repetitions). Finally, for any $\sigma \in \Sigma_X$, denote with $\mathcal{E}_\sigma$ the event that the set of the realized losses is $\sigma$. We have the following:

$$\max_{a \in [k]} \mathbb{E}\left[\sum_{t=1}^{T} \ell_t(a_t) - \ell_t(a)\right] = \max_{a \in [k]} \sum_{\sigma \in \Sigma_X} \mathbb{P}\left[\mathcal{E}_\sigma\right] \mathbb{E}\left[\sum_{t=1}^{T} \ell_t(a_t) - \ell_t(a)|\mathcal{E}_\sigma\right]$$

$$\leq \max_{a \in [k]} \mathbb{E}\left[\sum_{t=1}^{T} \ell_t(a_t) - \ell_t(a)|\mathcal{E}_{\hat{\sigma}}\right]$$

$$= \mathbb{E}\left[\max_{a \in [k]} \sum_{t=1}^{T} \ell_t(a_t) - \ell_t(a)|\mathcal{E}_{\hat{\sigma}}\right]$$

$$= R_T^{RO}(\mathcal{A}, \mathcal{S}_\pi),$$

where the first inequality follows by an averaging argument (i.e., there exists a $\hat{\sigma} \in \Sigma_X$ that dominates the average stochastic regret), and the last equality holds because once the multiset of realized values $\hat{\sigma}$ is fixed (we need multisets because there may be repeated elements), then all possible permutations are equally likely and thus we are back at the random-order model. Moreover, note that the $\max_{i \in [k]}$ only depends on the multiset $\hat{\sigma}$ and not on the specific realizations (so that we can safely move it into the expectation). Taking the sup with respect to the stochastic inputs yields the first inequality.

The argument for the second inequality is similar. Fix any random-order instance $\mathcal{S}$, characterized by a set of $T$ losses $\{h_1, \ldots, h_T\}$ that are randomly permuted according to $\pi$, and denote with $\mathcal{S}_\pi$ the corresponding adversarial input. Denote with $\Pi$ the set of all permutations over the $T$ rounds. We have the following:

$$\max_{a \in [k]} \mathbb{E}\left[\sum_{t=1}^{T} \ell_t(a_t) - \ell_t(a)\right] = \max_{a \in [k]} \frac{1}{T!} \sum_{\pi \in \Pi} \mathbb{E}\left[\sum_{t=1}^{T} h_{\pi(t)}(a_t) - h_{\pi(t)}(a)\right]$$

$$\leq \max_{\pi \in \Pi} R_T^{ADV}(\mathcal{A}, \mathcal{S}_\pi)$$

$$= R_T^{ADV}(\mathcal{A}, \mathcal{S}_{\pi^*}),$$

where $\pi^\star$ is the permutation that maximizes the regret. Taking the sup with respect to the random-order inputs yields the second inequality. $\square$

The above results tell that the minimax regret for the i.i.d. setting is at most that for the random-order one, which, in turn, is upper bounded by that of the adversarial input model.

## C. Concentration for Sampling without Replacement

In the original paper by Hoeffding (Hoeffding, 1963), a concentration bound for sampling without replacement is provided.

**Theorem C.1.** *Let $y_1, y_2, \ldots, y_T$ be a sequence of numbers in $[0,1]$, and consider an uniform random subset $S \subseteq \{y_1, \ldots, y_T\}$, where $s = |S|$. Then, the following inequality holds for any $\lambda > 0$:*

$$\mathbb{P}\left[\left|\frac{1}{s} \sum_{y_i \in S} y_i - \mu\right| \geq \lambda\right] \leq 2\exp(-s\lambda^2),$$

---

[13]Out of simplicity, we assume such support finite. The general statement can be proved with minimal arrangement.

where $\mu = \frac{1}{T}\sum_{t\in[T]} y_t$ is the average over the whole sequence.

Stated differently, if we have access to $s$ samples in the random-order model, we have that, with probability at least $(1-\delta)$, the following concentration holds:

$$\left|\frac{1}{s}\sum_{y_i\in S} y_i - \mu\right| \le \sqrt{\frac{\log(2/\delta)}{s}}. \tag{9}$$

For completeness, we note that random-order sequences typically concentrate *faster* than i.i.d. ones. There is, in fact, a refined analysis of Hoeffding inequalities that holds in the random-order model, due to Serfling (1974), that provides the following tighter concentration bound:

$$\left|\frac{1}{s}\sum_{y_i\in S} y_i - \mu\right| \le \sqrt{\left(1 - \frac{s-1}{T}\right)\frac{\log(2/\delta)}{2s}}. \tag{10}$$

## D. Missing Proof from Section 3

**Lemma 3.1.** *For any i.i.d. input and $T$ sufficiently large, it holds that $\mathbb{E}[\tau] \le 2\sqrt{T}$.*

*Proof.* It is immediate to observe that $\mathbb{E}[\tau]$ is maximized when the distribution of the losses on arm $1$ is the uniform distribution on the support $\{i/T, \text{ for } i = 1, 2, \ldots, T\}$, so we analyze that case. As a first step in the analysis, we find an integral formula for the expected value of $\tau$ (when $\ell_t(1)$ is drawn i.i.d. from $\{i/T, \text{ for } i = 1, 2, \ldots, T\}$), and then we study its asymptotic behaviour. We have the following chain of inequalities:

$$
\begin{aligned}
\mathbb{E}[\tau] &= \sum_{t=0}^{\infty} \mathbb{P}[\tau > t] \\
&= \sum_{t=0}^{T} \frac{T\cdots(T-t)}{T\cdots T} && \text{(Avoid collision up to time } t\text{)} \\
&= \sum_{t=0}^{T} \binom{T}{t}\frac{t!}{T^t} \\
&= \int_0^{+\infty} \sum_{t=0}^{T-1} \binom{T}{t}\left(\frac{x}{T}\right)^t e^{-x}dx && (\textstyle\int x^j e^{-x}dx = j!) \\
&= \int_0^{+\infty} \left(1 + \frac{x}{T}\right)^T e^{-x}dx \\
&= \sqrt{T}\int_0^{+\infty} \left(1 + \frac{y}{\sqrt{T}}\right)^T e^{-y\sqrt{T}}dy && (11)
\end{aligned}
$$

We then focus on the integral term, at the right-most side of Equation (11), and study the convergence of $\frac{\mathbb{E}[\tau]}{\sqrt{T}}$. Denote the integrand with $f_T(y)$, and consider its limit for large $T$:

$$\lim_{T\to+\infty} f_T(y) = \lim_{T\to+\infty} \exp\left[T\log\left(1 + \tfrac{y}{\sqrt{T}}\right) - y\sqrt{T}\right] = e^{-\frac{1}{2}y^2}.$$

Moreover, $f_T(y)$ is monotonically decreasing in $T$ and $f_1(y)$ is integrable; therefore, we can apply the dominated convergence theorem and argue that

$$\lim_{T\to+\infty} \frac{\mathbb{E}[\tau]}{\sqrt{T}} = \int_0^{+\infty} \lim_{T\to+\infty} f_T(y)dy = \int_0^{+\infty} e^{-\frac{1}{2}y^2}dy = \tfrac{1}{2}\sqrt{2\pi},$$

which concludes the proof. $\qquad\square$

## E. Missing Proof from Section 4.1

**Claim 4.3.** *For any block $i$, the corresponding clean event is realized with probability at least $1 - 1/T$.*

*Proof of Claim 4.3.* $\Delta_i(a)$ and $\Delta_i^p(a)$ have the same distribution, so we focus only on the former. By Hoedffding inequality for sampling without replacement (Theorem C.1), we have that, for each $a \in [k]$:

$$|\Delta_i(a) - \Delta(a)| \leq \sqrt{\frac{\log(2T)}{2^{i-1}}} \leq \epsilon_i$$

with probability at least $1 - \frac{1}{2T^2}$. The statement follows by union bounding over all $2k$ events (and $k \leq T$). $\qquad\square$

## F. Missing Proof from Section 4.2

**Claim 4.6.** *For all $i > i^\star$, under the event $\mathcal{E}_i$, we have that*

$$\mathrm{OPT}_{\mathrm{LP}} - \mathrm{OPT}_{\mathrm{LP}}^i \leq \epsilon_i \left(1 + \frac{3}{\rho}\right).$$

*Proof of Claim 4.6.* First, we consider the following inequalities:

$$\begin{aligned}
\mathrm{OPT}_{\mathrm{LP}} - \mathrm{OPT}_{\mathrm{LP}}^i &= \bar{r}^\top x^\star - r_i^{p,\top} x_i^\star \\
&= (\bar{r} - r_i^p)^\top x^\star + r_i^{p,\top}(x^\star - x_i^\star) \\
&\leq \epsilon_i + r_i^{p,\top}(x^\star - x_i^\star) \qquad\qquad \text{(Under } \mathcal{E}_i)
\end{aligned}$$

Now, we consider the second term. We will show that $x^\star$ is approximately feasible for the LP for which $x_i^\star$ is optimal and, thanks to the strict feasibility of action $\varnothing$, we obtain that $x^\star$ cannot be too much better than $x_i^\star$. Under the event $\mathcal{E}_i$ we have that

$$\bar{c}_j^{p,\top} x^\star \leq \bar{c}_j^\top x^\star + \epsilon_i \leq \rho + \epsilon_i,$$

which shows that $x^\star$ is $3\epsilon_i$-feasible for the constraints of $\mathrm{OPT}_{\mathrm{LP}}^i$. Then consider $\tilde{x} = \beta x^\star + (1 - \beta)\varnothing$ with $\beta = \frac{\rho - 2 \cdot \epsilon_i}{\rho + \epsilon_i}$, clearly $\bar{c}_j^{p,\top} \tilde{x} \leq \rho - \epsilon_i$ and thus it is feasible for $\mathrm{OPT}_{\mathrm{LP}}^i$ and $r_i^{p,\top} \tilde{x} = \beta \cdot r_i^{p,\top} x^\star$. Then

$$\begin{aligned}
r_i^{p,\top}(x^\star - x_i^\star) &= r_i^{p,\top}(\tilde{x} - x_i^\star) + r_i^{p,\top}(x^\star - \tilde{x}) \\
&\leq 0 + r_i^{p,\top}(x^\star - \tilde{x}) \qquad\qquad (\tilde{x} \text{ is feas. } \mathrm{OPT}_{\mathrm{LP}}^i) \\
&\leq (1 - \beta)r_i^{p,\top} x^\star \leq \frac{3\epsilon_i}{\rho} \qquad\qquad (\|r_i^p\|_\infty \leq 1)
\end{aligned}$$

which concludes the proof. $\qquad\square$

## G. Missing Proofs from Section 4.3

We start analyzing the algorithm by counting the number of switches. Regardless of the instance and the randomness of the algorithm, it switches action at most $k + 1$ times in each of the $O(\log T)$ time blocks. This is summarized in the following Lemma.

**Lemma G.1** (Low-Switching). *For any realization of* SIMULATION-SUCCESSIVE-ELIMINATION, *the total switching cost is at most $2k \log T$.*

We now restate and prove the regret guarantees of SIMULATION-SUCCESSIVE-ELIMINATION.

**Theorem 4.9.** *There exists an algorithm for bandits with switching costs in the random-order model with regret $O(\sqrt{kT \log^3 T})$.*

*Proof.* Fix any random-order instance $\mathcal{S}$, on a set of $T$ vectors $\{h_1, \ldots, h_T\}$, and denote with $\ell(a)$ the average loss associated with each action $a$: $\ell(a) = \sum_{t \in [T]} \ell_t(a)/T$. Denote with $a^\star$ any action with maximum $\ell$, and introduce the gap $\Delta(a) = \ell(a) - \ell(a^\star)$. For any time block $i$, let $\hat{\ell}_i(a)$ denote the value of the estimator of $\hat{\ell}(a)$ at the end of that time block, and denote with $N_i(a)$ the number of blocks (included the $i^{th}$) in which action $a$ has been active. Note that once an action is deactivated, it stays like this for the rest of the algorithm.

For each time block $T_i$, there are at most $k$ ways in which it can be partitioned into the sub-blocks used by the round-robin phase of the algorithm, depending on the cardinality of the active set $A$. We make here simplifying assumptions for the calculations: we assume that the time blocks are perfectly divisible into sub-blocks. This is to avoid considering the suffix of the time block corresponding to the remainder of the division in sub-blocks. Note, this assumption is to simplify calculations: having more samples regarding the last active action only improves concentration.

For each action $a$ and block $i$, we define the following clean event that requires that $\ell(a)$ is well estimated across all possible sub-blocks. In block $i$ we denote with $B_i^j(g)$ the $g^{th}$ sub-block of $2^i/j$ time steps, i.e., $B_i^j(g) = \{2^i(1 + g/j) + 1, \ldots, 2^i(1 + (g+1)/j)\}$:

$$\mathcal{E}_i^a = \left\{ \left| \sum_{t \in B_i^j(g)} \frac{\ell_t(a)}{2^i/j} - \ell(a) \right| \leq \sqrt{10 \frac{\log T}{2^i/j}} \quad \forall j \in [k] \text{ and } g \in [j] \right\}$$

**Claim G.2.** *For any block $i$ and action $a$, the clean even $\mathcal{E}_i^a$ has probability at least $1 - 1/T^3$.*

*Proof.* Fix any $B_i^j(g)$, by applying Hoeffding for sampling with replacement (Theorem C.1), we get that the following inequality holds with probability at least $1 - T^{-5}$ :

$$\left| \sum_{t \in B_i^j(g)} \frac{\ell_t(a)}{2^i/j} - \ell(a) \right| \leq \sqrt{10 \frac{\log T}{2^i/j}}$$

Union bounding over all $k$ choices of $j$ and (at most) $k$ choices of $g$ yields the desired statement. □

It is now natural to look at the overall clean event $\mathcal{E}$ by intersecting the $\mathcal{E}_i^a$ over all $\log T$ time blocks and $k$ actions. By union bounding, we hence get that the clean event holds with high probability.

**Claim G.3** (Clean event)**.** *The clean event has probability at least $1 - 1/T$.*

Before proceeding, we underline that the clean event does not depend in any way on the algorithm, but is only a probabilistic statement over the random permutation, which considers *fixed* intervals. The crucial observation is that, under the clean event, we are sure that SIMULATION-SUCCESSIVE-ELIMINATION maintains precise estimates throughout. More precisely, denote with $\hat{\ell}_i(a)$ the estimate of $\ell(a)$ maintained at the end of block $i$, we have the following claim:

**Claim G.4.** *For all block $i$ and active action $a$, conditioning on the clean event, we have that*

$$|\hat{\ell}_i(a) - \ell(a)| \leq \sqrt{5k \frac{\log^3 T}{2^i}}$$

*Proof.* If the action $a$ is active at the beginning of block $i$, then it has been explored in all previous blocks. Denote with $T(a)$ the number of times action $a$ has been played up to block $i$ included, and with $T_j(a)$ the times in the generic block $T_j$ when action $a$ has been played; clearly $|T_j|$ is at least $2^j/k$ and $T(a) = \sum_{j \leq i} T_i(a)$.

We have the following result:

$$
\begin{aligned}
\left| \hat{\ell}_i(a) - \ell(a) \right| &= \left| \frac{1}{T(a)} \sum_{j \leq i} \sum_{t \in T_j(a)} \ell_t(a) - \ell(a) \right| \\
&\leq \frac{1}{T(a)} \sum_{j \leq i} T_j(a) \left| \sum_{t \in T_j(a)} \frac{\ell_t(a)}{T_j(a)} - \ell(a) \right| \\
&\leq \frac{1}{T(a)} \sum_{j \leq i} T_j(a) \sqrt{10 \frac{\log T}{T_j(a)}} && \text{(By definition of clean event)} \\
&= \frac{1}{T(a)} \sum_{j \leq i} \sqrt{10 \cdot T_j(a) \log T} \\
&\leq \sqrt{10 \frac{\log^3 T}{T(a)}} && \text{(By Jensen Inequality)} \\
&\leq \sqrt{5k \frac{\log^3 T}{2^i}},
\end{aligned}
$$

where in the last inequality we used that an active action is played with frequency at least $1/k$. $\qquad\square$

The fact that all active actions are well estimated (and that there always exists at least one active action) implies that the best action $a^\star$ is always active, under the clean event. In fact, for any block $i$ and active action $a$ it holds that

$$
\hat{\ell}_i(a^\star) - \epsilon_i \leq \ell(a^\star) \leq \ell(a) \leq \hat{\ell}(a) + \epsilon_i,
$$

which contradicts the deactivation rule. We have then proved that the instantaneous regret in the generic block $i$, under the clean event, is at most $2\epsilon_i$. We know that the clean event is realized with high probability (Claim G.3), and that the number of switches is $O(k \log T)$ (Lemma G.1). All in all, we have the following bound on the regret:

$$
R_T(\text{SSE}) \leq 2k \log T + 4 \sum_{i=1}^{\log T} 2^i \epsilon_i \in O(\sqrt{kT \log^3 T}),
$$

thus concluding the proof. $\qquad\square$

## H. Classification in the Random-Order Input Model

We study the classical model in binary classification: we have a family $\mathcal{H}$ of hypotheses $h$ defined on a set $\mathcal{X}$ for which a bounded loss function $\ell$ is defined. Formally, for any hypothesis $h$ and pair $(x, y)$ with $x \in \mathcal{X}$ and $y \in \{0, 1\}$, $\ell(h(x), y) \in [0, 1]$ denotes the cost incurred by predicting label $h(x)$ on $(x, y)$ (see , e.g., Bousquet et al., 2003). Note, to be uniform with the literature, in this section we use $\ell$ to denote the loss function (as opposed to the loss vectors as in the rest of the paper) and $h$ for classifiers (as opposed to the non-shuffled input loss vectors as in the rest of the paper).

**PAC Model.** In the PAC learning model, the learner has access to i.i.d. samples from a joint distribution $\mathcal{D}$ from which pairs $(X, Y)$ are drawn, and its goal is to quickly estimate or approximate the best classifier, i.e., the classifier that minimizes $\mathbb{E}[\ell(h(X), Y)]$. It is a standard result in learning theory that a class $\mathcal{H}$ is learnable, i.e., the best classifier is arbitrarily approximable if and only if the $d_{\text{VC}}$ dimension of $\mathcal{H}$ is bounded, and that the convergence rate also depends on $d_{\text{VC}}$ (see, e.g., Vapnik & Chervonenkis, 1971).

**(Adversarial) Online Model.** In the online model (Littlestone, 1987), the input-label pairs are generated adversarially, and the performance of a learning algorithm is measured with respect to the best hypothesis in $\mathcal{H}$. Surprisingly, the online learnability of a family $\mathcal{H}$ is governed by a stricter notion of dimension, the Littlestone dimension.

**Random-Order Model.** In this paper, we study the random-order model: the input is provided by a multiset $\mathcal{S}$ of $(x, y)$ pairs, which are presented by the learner in uniform random order. The performance measure of a learning algorithm is its regret with respect to the best classifier on $\mathcal{S}$. As the input arrives according to some random permutation $\pi$, we let $\ell_t^\pi(h) = \ell(h(x_{\pi(t)}), y_{\pi(t)})$ be the loss of hypothesis $h \in \mathcal{H}$ at time $\pi(t)$. Similarly, we let $\hat{\ell}_t^\pi(h) = \frac{1}{t} \cdot \sum_{\tau:\pi(\tau)\leq\pi(t)} \ell_\tau^\pi(h)$ be the empirical loss accrued by time $\pi(t)$, and $\hat{\ell}_{t:T}^\pi(h) = \frac{1}{T-t+1} \cdot \sum_{\tau:\pi(\tau)>\pi(t)} \ell_\tau^\pi(h)$. Moreover, we denote by $\bar{\ell}(h) = \frac{1}{T} \cdot \sum_{\tau=1}^{T} \ell(h(x_\tau), y_\tau)$ the average loss of hypothesis $h$ on the whole dataset composed of $T$ input-label pairs.

## H.1. Sample Complexity with Random-Order Input

**Lemma H.1.** *Consider the problem of online binary classification in the random-order model. For a hypothesis class $\mathcal{H}$ with VC-dimension $d_{\mathrm{VC}}$, given the first $t-1$ out of $T$ samples from a uniform permutation $\pi$, it holds that, for all $h \in \mathcal{H}$,*

$$|\bar{\ell}(h) - \hat{\ell}_{t-1}^\pi(h)| \leq \frac{1}{2} \cdot \left( \sqrt{\frac{8d_{\mathrm{VC}}}{t-1} \log\left(\frac{2e(t-1)}{d_{\mathrm{VC}}}\right)} + \sqrt{\frac{8}{t-1} \log\left(\frac{2}{\delta}\right)} \right),$$

*with probability at least $1 - \delta$.*

The proof of Lemma H.1 implements a version of the classical "symmetrization" argument for uniform convergence in the i.i.d. setting (Bousquet et al., 2003). We consider a "ghost sample" made of $t-1$ out of $T$ samples, that is the set of input-label pairs $(x_{\pi'(\tau)}, y_{\pi'(\tau)})_{\tau:\pi(\tau)<\pi(t)}$ extracted according to an independent uniform permutation $\pi'$.

**Claim H.2.** *Fix any time $t > 1$ and precision $\epsilon_t > 0$ such that $(t-1)\epsilon_t^2 \geq 8$, then the following inequality holds:*

$$\mathbb{P}_\pi\left[\sup_{h\in\mathcal{H}} |\bar{\ell}(h) - \hat{\ell}_{t-1}^\pi(h)| \geq \epsilon_t\right] \leq 2\mathbb{P}_{\pi,\pi'}\left[\sup_{h\in\mathcal{H}} |\hat{\ell}_{t-1}^\pi(h) - \hat{\ell}_{t-1}^{\pi'}(h)| \geq \frac{\epsilon_t}{2}\right]. \tag{12}$$

*Proof.* For any realization of $\pi$ and $\pi'$, and any classifier $h \in \mathcal{H}$, we have that

$$\mathbb{I}_{\left\{|\bar{\ell}(h)-\hat{\ell}_{t-1}^\pi(h)|>\epsilon_t\right\}}\mathbb{I}_{\left\{|\bar{\ell}(h)-\hat{\ell}_{t-1}^{\pi'}(h)|<\frac{\epsilon_t}{2}\right\}} \leq \mathbb{I}_{\left\{|\hat{\ell}_{t-1}^\pi(h)-\hat{\ell}_{t-1}^{\pi'}(h)|>\frac{\epsilon_t}{2}\right\}},$$

since for any three reals $a, b, c$ and $\epsilon > 0$, if $|a-b| > \epsilon$ and $|a-c| < \frac{\epsilon}{2}$, then

$$|a-b| \leq |a-c| + |b-c| \implies |b-c| > \frac{\epsilon}{2}.$$

We take expectations with respect to the second sample extracted according to $\pi'$ and get

$$\mathbb{I}_{\left\{|\bar{\ell}(h)-\hat{\ell}_{t-1}^\pi(h)|>\epsilon_t\right\}}\mathbb{P}_{\pi'}\left[|\bar{\ell}(h) - \hat{\ell}_{t-1}^{\pi'}(h)| < \frac{\epsilon_t}{2}\right] \leq \mathbb{P}_{\pi'}\left[|\hat{\ell}_{t-1}^\pi(h) - \hat{\ell}_{t-1}^{\pi'}(h)| > \frac{\epsilon_t}{2}\right].$$

Since $\bar{\ell}(h) = \mathbb{E}_{\pi'}\left[\hat{\ell}_{t-1}^{\pi'}(h)\right]$ for all $h \in \mathcal{H}$ and $t > 1$, we can apply Chebyšev's inequality and obtain

$$\mathbb{P}_{\pi'}\left[|\bar{\ell}(h) - \hat{\ell}_{t-1}^{\pi'}(h)| \geq \frac{\epsilon_t}{2}\right] \leq \frac{4\mathbb{E}_{\pi'}[(\hat{\ell}_{t-1}^{\pi'}(h))^2]}{\epsilon_t^2} \leq \frac{4(t-1)}{(t-1)^2\epsilon_t^2} = \frac{4}{(t-1)\epsilon_t^2}.$$

The second inequality follows because the second moment is bounded in $[0,1]$, as losses are bounded in $[0,1]$. Therefore,

$$\mathbb{I}_{\left\{|\bar{\ell}(h)-\hat{\ell}_{t-1}^\pi(h)|>\epsilon_t\right\}}\left(1 - \frac{4}{(t-1)\epsilon_t^2}\right) \leq \mathbb{P}_{\pi'}\left[|\hat{\ell}_{t-1}^\pi(h) - \hat{\ell}_{t-1}^{\pi'}(h)| > \frac{\epsilon_t}{2}\right] \leq \mathbb{P}_{\pi'}\left[\sup_{h\in\mathcal{H}} |\hat{\ell}_{t-1}^\pi(h) - \hat{\ell}_{t-1}^{\pi'}(h)| > \frac{\epsilon_t}{2}\right].$$

We can now apply the sup over all $h \in \mathcal{H}$ on the left-hand side of the above inequality and then take the expectation also with respect to the randomness of $\pi$. This implies the statement, as $(t-1)\epsilon_t^2 \geq 8$. $\square$

*Proof of Lemma H.1.* Claim H.2 allows us to replace $\bar{\ell}(h)$ by an empirical average over the ghost sample. As a result, the right-hand side only depends on the projection of the class $\mathcal{H}$ on the double sample drawn from $\pi, \pi'$, $\mathcal{H}\left((x_{\pi(\tau)}, y_{\pi(\tau)})_{\tau\in[t]}, (x_{\pi'(\tau)}, y_{\pi'(\tau)})_{\tau\in[t]}\right)$, denoted succinctly as $\mathcal{H}_t^{\pi,\pi'}$, which gives

$$\mathbb{P}_{\pi,\pi'}\left[\sup_{h\in\mathcal{H}} |\hat{\ell}_{t-1}^\pi(h) - \hat{\ell}_{t-1}^{\pi'}(h)| \geq \frac{\epsilon_t}{2}\right] = \mathbb{P}_{\pi,\pi'}\left[\sup_{h\in\mathcal{H}_t^{\pi,\pi'}} |\hat{\ell}_{t-1}^\pi(h) - \hat{\ell}_{t-1}^{\pi'}(h)| \geq \frac{\epsilon_t}{2}\right].$$

We now apply Hoeffding's inequality (as in Theorem C.1) on $\hat{\ell}_{t-1}^{\pi}(h) - \hat{\ell}_{t-1}^{\pi'}(h)$, for a fixed $h \in \mathcal{H}$. To see why it is possible, we construct an equivalent process to extracting a sample and a "ghost sample" according to $\pi$ and $\pi'$ respectively. Consider losses presented in an arbitrary ordering $\ell_1(h), \ldots, \ell_T(h)$ and an ordering induced by the uniform random permutation $\pi'$, $\ell_1^{\pi'}(h), \ldots, \ell_T^{\pi'}(h)$. Let $w_\tau(h) = (\ell_\tau(h), \ell_\tau^{\pi'}(h))$ be the pair of variables matched by index, $f(w_\tau(h)) = \ell_\tau(h) - \ell_\tau^{\pi'}(h)$, and $\mathcal{F}(h) = \{f(w_\tau(h)) \mid \tau \in [T]\}$. Note that, for every $\pi'$,

$$\sum_{\tau \in [T]} f(w_\tau(h)) = \sum_{\tau \in [T]} \ell_\tau(h) - \sum_{\tau \in [T]} \ell_\tau^{\pi'}(h) = 0.$$

We extract a uniform random subset $F$ composed of $t-1$ functions $f(w_\tau(h))$ from $\mathcal{F}(h)$, and get

$$\mathbb{P}_{F \sim \mathcal{F}(h)} \left[ \left| \frac{1}{t-1} \sum_{\tau \in F} f(w_\tau(h)) \right| \geq \epsilon_t \right] \leq 2 \exp\left(-(t-1)\epsilon_t^2\right).$$

by Hoeffding's inequality (as in Theorem C.1). Therefore, we have

$$\mathbb{P}_{\pi,\pi'} \left[ |\hat{\ell}_{t-1}^{\pi}(h) - \hat{\ell}_{t-1}^{\pi'}(h)| \geq \frac{\epsilon_t}{2} \right] \leq 2 \exp\left(-\frac{(t-1)\epsilon_t^2}{4}\right). \tag{13}$$

Combining everything, we obtain

$$\mathbb{P}_\pi \left[ \sup_{h \in \mathcal{H}} |\bar{\ell}(h) - \hat{\ell}_{t-1}^{\pi}(h)| \geq \epsilon_t \right] \leq 2\mathbb{P}_{\pi,\pi'} \left[ \sup_{h \in \mathcal{H}} |\hat{\ell}_{t-1}^{\pi}(h) - \hat{\ell}_{t-1}^{\pi'}(h)| \geq \frac{\epsilon_t}{2} \right]$$

$$= 2\mathbb{P}_{\pi,\pi'} \left[ \sup_{h \in \mathcal{H}_t^{\pi,\pi'}} |\hat{\ell}_{t-1}^{\pi}(h) - \hat{\ell}_{t-1}^{\pi'}(h)| \geq \frac{\epsilon_t}{2} \right]$$

$$\leq 2 \cdot \left(\frac{e(t-1)}{d_{\mathrm{VC}}}\right)^{d_{\mathrm{VC}}} \cdot \mathbb{P}_{\pi,\pi'} \left[ |\hat{\ell}_{t-1}^{\pi}(h) - \hat{\ell}_{t-1}^{\pi'}(h)| \geq \frac{\epsilon_t}{2} \right]$$

$$\leq 4 \cdot \left(\frac{e(t-1)}{d_{\mathrm{VC}}}\right)^{d_{\mathrm{VC}}} \cdot \exp\left(-\frac{(t-1)\epsilon_t^2}{4}\right).$$

The second inequality is an application of the Sauer-Shelah lemma (Vapnik & Chervonenkis, 1971; Sauer, 1972; Shelah, 1972), and the last by the expression in (13). The proof concludes by setting

$$\epsilon_t = \frac{1}{2} \cdot \left( \sqrt{\frac{8 d_{\mathrm{VC}}}{t-1} \log\left(\frac{2e(t-1)}{d_{\mathrm{VC}}}\right)} + \sqrt{\frac{8}{t-1} \log\left(\frac{2}{\delta}\right)} \right). \qquad \square$$

## H.2. Regret of an Empirical Risk Minimizer with Random-Order Input

**Theorem H.3.** *Consider the problem of online binary classification in the random-order model. Fo a hypothesis class $\mathcal{H}$ with VC-dimension $d_{\mathrm{VC}}$, given the first $t-1$ out of $T$ samples from a uniform permutation $\pi$, the algorithm $\mathcal{A}^{\mathrm{RO}}$ that returns the minimizer*

$$h_t \in \arg\min_{h \in \mathcal{H}} \sum_{\tau : \pi(\tau) < \pi(t)} \ell_\tau^{\pi}(h),$$

*achieves a regret of at most*

$$R_T(\mathcal{A}^{\mathrm{RO}}, \mathcal{S}) \leq 8 \sqrt{d_{\mathrm{VC}} T \log\left(\frac{T}{d_{\mathrm{VC}}}\right)}.$$

*Proof.* Define, for any $t \in [T]$, the clean event in the past as $\mathcal{E}_t^{\mathrm{past}} = \{\pi \mid |\bar{\ell}(h) - \hat{\ell}_{t-1}^{\pi}(h)| \leq \epsilon_t \ \forall h \in \mathcal{H}\}$ and in the future as $\mathcal{E}_t^{\mathrm{future}} = \{\pi \mid |\bar{\ell}(h) - \hat{\ell}_{t:T}^{\pi}(h)| \leq \epsilon_{T-t+1} \ \forall h \in \mathcal{H}\}$. The clean event at time $t$ is $\mathcal{E}_t = \mathcal{E}_t^{\mathrm{past}} \cap \mathcal{E}_t^{\mathrm{future}}$.

The instantaneous regret at time $\pi(t)$ under the clean event is

$$
\begin{aligned}
\ell_t^\pi(h_t) - \ell_t^\pi(h^*) &\leq \ell_t^\pi(h_t) - \hat{\ell}_{t-1}^\pi(h_t) + \hat{\ell}_{t-1}^\pi(h^*) - \ell_t^\pi(h^*) \\
&= (\ell_t^\pi(h_t) - \bar{\ell}(h_t)) + (\bar{\ell}(h_t) - \hat{\ell}_{t-1}^\pi(h_t)) + (\hat{\ell}_{t-1}^\pi(h^*) - \bar{\ell}(h^*)) + (\bar{\ell}(h^*) - \ell_t^\pi(h^*)) \\
&\leq (\ell_t^\pi(h_t) - \bar{\ell}(h_t)) + (\ell_t^\pi(h^*) - \bar{\ell}(h^*)) + 2\epsilon_t.
\end{aligned}
$$

The first inequality above follows by definition of $h_t$, the empirical loss minimizer, which gives $\hat{\ell}_{t-1}^\pi(h^*) - \hat{\ell}_{t-1}^\pi(h_t) \geq 0$. Note that the clean event on the order in which the elements arrive but only on the following two multisets: $\mathcal{S}^{\text{past}}, \mathcal{S}^{\text{future}}$, i.e., the multisets that represent the past (from time $\pi(1)$ until $\pi(t-1)$) and future (from time $\pi(t)$ until $\pi(T)$). We now take the expectation conditioning on the clean event and the two multisets $\mathcal{S}^{\text{past}}, \mathcal{S}^{\text{future}}$. Therefore,

$$
\begin{aligned}
\mathbb{E}\left[\ell_t^\pi(h_t) - \ell_t^\pi(h^*) \mid \mathcal{E}_t, \mathcal{S}^{\text{past}}, \mathcal{S}^{\text{future}}\right] &\leq 2\epsilon_t + \mathbb{E}\left[(\ell_t^\pi(h_t) - \bar{\ell}(h_t)) + (\ell_t^\pi(h^*) - \bar{\ell}(h^*)) \mid \mathcal{E}_t, \mathcal{S}^{\text{past}}, \mathcal{S}^{\text{future}}\right] \\
&= 2\epsilon_t + \sum_{(x,y) \in \mathcal{S}^{\text{future}}} \frac{1}{T-t+1} \cdot \mathbb{E}\left[(\ell(h_t(x), y) - \bar{\ell}(h_t)) + (\ell(h^*(x), y) - \bar{\ell}(h^*)) \mid \mathcal{E}_t, \mathcal{S}^{\text{past}}, \mathcal{S}^{\text{future}}\right] \\
&\leq 2\epsilon_t + 2 \cdot \mathbb{E}\left[\sup_{h \in H} |\bar{\ell}(h) - \hat{\ell}_{t:T}^\pi(h)| \mid \mathcal{E}_t, \mathcal{S}^{\text{past}}, \mathcal{S}^{\text{future}}\right] \\
&\leq 2(\epsilon_t + \epsilon_{T-t+1}).
\end{aligned}
$$

The equality above holds because once $\mathcal{S}^{\text{past}}, \mathcal{S}^{\text{future}}$ are fixed, the loss at time $\pi(t)$ is uniformly sampled from the future. By the tower property of conditional expectation (on all the possible values of $\mathcal{S}^{\text{past}}, \mathcal{S}^{\text{future}}$), we have proven that the expected instantaneous regret conditioned on the clean event $\mathcal{E}_t$ is at most:

$$
\mathbb{E}\left[\ell_t^\pi(h_t) - \ell_t^\pi(h^*) \mid \mathcal{E}_t\right] \leq 2(\epsilon_t + \epsilon_{T-t+1}) \leq \sqrt{32 d_{\text{VC}} \log\left(\frac{2eT}{d_{\text{VC}}}\right)} \cdot \left(\sqrt{\frac{1}{t-1}} + \sqrt{\frac{1}{T-t+1}}\right),
$$

where the last inequality holds by taking $\delta = 1/T$ in Lemma H.1. By definition of regret and under this choice of $\delta$, we have

$$
\begin{aligned}
R_T(\mathcal{A}^{\text{RO}}, \mathcal{S}) &= \sum_{t \in [T]} \mathbb{E}\left[\ell_t^\pi(h_t) - \ell_t^\pi(h^*) \mid \mathcal{E}_t\right] \cdot \mathbb{P}\left[\mathcal{E}_t\right] + \mathbb{E}\left[\ell_t^\pi(h_t) - \ell_t^\pi(h^*) \mid \bar{\mathcal{E}}_t\right] \cdot \mathbb{P}\left[\bar{\mathcal{E}}_t\right] \\
&\leq \sqrt{32 d_{\text{VC}} \log\left(\frac{2eT}{d_{\text{VC}}}\right)} \cdot \sum_{t > 1} \left(\sqrt{\frac{1}{t-1}} + \sqrt{\frac{1}{T-t+1}}\right) + T \cdot \frac{1}{T} \\
&\leq 8\sqrt{d_{\text{VC}} T \log\left(\frac{T}{d_{\text{VC}}}\right)},
\end{aligned}
$$

which concludes the proof. $\qquad\square$

