# OpenReview forum: "Online Learning in the Random-Order Model"
_ICML.cc/2025/Conference — ICML 2025 poster_

### Official Review · Reviewer_xKSP · 2025-03-07

**Overall Recommendation:** 3

**Summary:**

This paper studies online learning in the random order model, where the loss functions are fixed in advance but are presented in a random order. It has been known that the random order model lies between the stochastic model, where the loss functions are drawn from a fixed distribution, and the adversarial model, where loss functions are selected adversarially. This paper focuses on designing algorithms under the random order model by making use of algorithms under the stochastic models. On the one hand, the paper gives an example that shows naively running an algorithm that works under the stochastic model would result in a linear regret under the random order model. On the other hand, this paper introduces a framework that designs no-regret learning algorithms under the random order model using no-regret learning algorithms under the stochastic model. Examples of how this framework works include Prediction with Delayed Feedback and Online Learning with Constraints.

**Claims And Evidence:**

The claims and related technical lemmas are stated clearly and the proofs of the statements are clear.

**Essential References Not Discussed:**

The setting studied in this paper is new and I think there are not many prior works closely related to this paper.

**Experimental Designs Or Analyses:**

This paper is theoretical and does not need experiments to support the results.

**Methods And Evaluation Criteria:**

This paper uses regret bounds as a measurement of the performance of the online learning algorithms, which is standard in analyzing online learning algorithms.

**Other Comments Or Suggestions:**

Though not directly related, models such as online learning in the best order/ worst order/self-directed order models have been proposed since the 1990s. It could be helpful if related papers such as

*Ben-David, Shai, Eyal Kushilevitz, and Yishay Mansour. "Online learning versus offline learning." Machine Learning 29 (1997): 45-63.*
can be cited and discussed.

**Other Strengths And Weaknesses:**

Strengths:
1. Random order is an intermediate model that lies between the stochastic setting and the adversarial setting. This paper shows that under the finite time analysis, the random order model is different from the stochastic setting and also designs an algorithmic framework that can convert existing no-regret learning algorithms in the stochastic setting into a no-regret learning algorithm under the random order model. This framework might be useful for studying other online learning problems in the random order model
2. The statements and the proofs of the theorems and the technical lemmas are clear. This makes the paper easy to follow.

Weakness:
I am confused about the structure of the main part of the paper. It is a bit unclear what the central result the paper wants to sell as there are many different online learning problems considered in this paper. Furthermore, it seems that bandits with switching costs as well as classification in the random order model are both important parts of this paper. However, even informal statements of results from these parts are not stated in the main body. In particular, it seems that the results on classification in the random order do not have deep connections with the simulation framework. These make the structure of the main part of the paper strange and confusing.

**Questions For Authors:**

I am very confused about the motivation for analyzing the birthday testing problem in the stochastic setting. The first phase of the birthday testing problem looks quite artificial, even without the first phase, the algorithm should still be no-regret and by removing the first phase it should not result in a linear regret under the random order model. So, I am not sure what the analysis of the birthday testing problem implies.

**Relation To Broader Scientific Literature:**

Online learning with adversarial order and stochastic order has been studied extensively. The random order model though is asymptotically equivalent to the stochastic setting, is different from the stochastic setting in finite time. The algorithmic framework presented in this paper might inspire designing no-regret learning algorithms for other online learning problems under the random order model.

**Theoretical Claims:**

I only checked the proofs of the statements in the main body of the paper, For theorems in the appendix, I only checked the statements of the technical lemmas used for proving the theorems but did not check the proof carefully.

---

> ### Author Rebuttal · Authors · 2025-04-01
>
> We thank the reviewer for the positive feedback about the paper.
>
> **Importance of the BIRTHDAY-TEST.** In Section 3, we prove that there is no black-box reduction from random-order to i.i.d. That is, we cannot hope to prove a statement of the form: “Any algorithm with sublinear regret in the stochastic setting has sublinear regret in the RO instance” which, in principle, could reasonably be expected. To argue about this impossibility, we exhibit a specific algorithm (BIRTHDAY-TEST) that is no-regret against any stochastic i.i.d. instance but fails against a specific random-order instance. We stress that this algorithm is only used in the proof of this impossibility result.
>
> **Structure of the paper.** Unfortunately, we had to move important parts of our results to the appendix in the interest of space. We will exploit the extra page in the camera-ready version (and possibly move/shorten Section 3) to move parts of Appendices E and F in the main body.
>
> **Ben-David et al. 97.** We thank the reviewer for pointing out this paper. Although not directly related to our model (we study random order, and they study online, worst-case offline, best-case offline, and self-directed sequences), they share our interest in investigating alternative input generation models beyond adversarial and i.i.d. We will add a discussion in the camera-ready version.

---

### Official Review · Reviewer_82fy · 2025-03-07

**Overall Recommendation:** 3

**Summary:**

- This paper studies online learning in the random order model. Here, the adversary can pick an arbitrary sequence of loss functions, but must randomly permute them before presenting them one at a time to the learning algorithm.
- They show that, in full generality, online algorithms with low-regret under stochastic adversaries fail to obtain sublinear regret under random-order adversaries.
-  On the other hand, the authors show how to use an online learning algorithm for stochastic adversaries to construct an online learning algorithm under a random-order adversary with minimal blow up in regret guarantees.
- Using this conversion, they give improved regret bounds for several settings: prediction with delayed feedback, online learning with constraints, bandits with switching costs, and online classification.
- For online classification, they show that the VC dimension is sufficient for learnability under random-order adversaries.

## Update after rebuttal

I thank the authors for their response. As they have satisfactorily addressed my questions and concerns, I will maintain my positive score for this paper.

**Claims And Evidence:**

To me, the claims made in the submission are supported by clear and convincing evidence.

**Essential References Not Discussed:**

With regards to online classification, a recent paper [1] studies a different type of intermediate setting where predictions about the future examples that need to be classified is available to the learner. Similar to the findings of this paper, the authors there show that for binary classification, VC dimension is also sufficient when such predictions are available.

[1] Raman, Vinod, and Ambuj Tewari. "Online Classification with Predictions." NeurIPS (2024).

**Experimental Designs Or Analyses:**

This is a purely theoretical paper with no experiments.

**Methods And Evaluation Criteria:**

Yes, the authors evaluate their algorithms using the standard notion of regret.

**Other Comments Or Suggestions:**

I would move the sentence "Let $n_i(a)$ denote the times..." under (iii)  to after the sentence "Run algorithm $A$..." in (ii) in Simulation Procedure. It was not immediately clear to me that $n_i(a)$ was the number of times $A$ played action $a$ on iid data from $D_i$.

**Other Strengths And Weaknesses:**

**Strengths:**
- The paper is well-written and easy to follow.
- The conversion is intuitive and natural

**Weaknesses:**
One gripe I have with this paper is with its organization.
- In Section 1.1, the authors state that they apply their conversion to get improved regret bounds for 4 settings: prediction with delayed feedback, online learning with constraints, bandits with switching costs, and online classification. Yet, in the main text, only two of these are discussed in detail. My suggestion would be to move Section 3 to the Appendix and use the extra space to provide more details about your results for bandits with switching costs and online classification.
- In addition, the authors do not provide a comparison between the regret bounds they achieved under a random-order adversaries ad the optimal regret bounds under a worst-case adversaries. This makes it hard to tell how much of a quantitative advantage a random order provides. It would be helpful to have a table summarizing your results for the random-order adversary as well as the optimal regret bounds for a worst-case adversary.

My other issue with this paper is that the results are not very surprising as they just result from running the stochastic algorithm on sub-samples of the history. This approach is intuitive, but it is not clear that it is optimal for the random-order model in any of the settings that the authors study. Unfortunately, the authors do not provide lower bounds for the random-order adversaries in any of their settings.

**Questions For Authors:**

(1) What are the lower bounds for random order adversaries in each of the settings you study? If the stochastic algorithm is optimally chosen, do you obtain the optimal regret bounds under random order adversaries in any of your settings?

**Relation To Broader Scientific Literature:**

This paper fits nicely into the recent interest of going beyond worst-case adversaries in online learning. Here, the goal is to obtain better regret guarantees by weakening the adversary in some reasonable way. From this perspective, this paper shows that what makes online learning hard under a worst-case adversary is not the fact that the adversary can pick an arbitrary sequence of loss functions, but the order of these loss functions. Indeed, this paper shows that even if you allow the adversary to pick the set of loss functions arbitrarily, online learning can be as easy as batch/stochastic learning if the adversary cannot control the order in which they are shown to the learner.

**Theoretical Claims:**

I went through the proofs of Section 4.1 in the main text. The claims seem to check out to me.

---

> ### Author Rebuttal · Authors · 2025-04-01
>
> We thank the reviewer for their feedback and positive feedback about the paper.
>
> **Paper organization.** Unfortunately, we had to move part of our results to the appendix in the interest of space. We thank the reviewer for the suggestion, and we will exploit the extra page in the camera ready (and move/shorten Section 3) to move part of Appendices E and F in the main body.
>
> **Tightness of our results.** We would like to point out to the reviewer that any lower bound for the stochastic setting also applies to the random order one (see also Appendix B); therefore, our results are tight (up to poly-log terms) for the random-order model (as they match the stochastic lower bounds). We highlight that all the settings of interest had minimax rates in the adversarial setting strictly worse than in the stochastic one (see section 1.1). For example, in bandits with switching costs, the adversarial setting is $T^{2/3}$ minimax optimal, while in the RO (and stochastic) is $\sqrt{T}$ minimax optimal. We thank the reviewer for the idea of a table summarizing the result. We will implement it in the camera-ready.
>
> **Comparison between random order and stochastic models.** While we expected the RO model to be somewhat closer to the stochastic model than the adversarial one, we strongly believe quantifying the “distances” to be non-obvious.
> In particular, our construction proving the non-existence of a black-box reduction from the stochastic to the random-order model shows that any similar “conversion result” needs a non-trivial component. We find the training and testing procedure both intuitive and interesting, as it successfully distills the similarities between random-order and i.i.d. instances. Finally, we find it surprising that the results we obtain are also tight in all the settings we consider (see the answer to Tightness of our results). We thank the reviewer who pointed out that this should be highlighted better. We will add a discussion in the final version.
>
> **Additional literature.** We thank the reviewer for pointing out the recent NeurIPS’24 paper [1]. We will discuss it in the camera-ready.
>
> **Other comments and suggestions.** We will move the sentence about $n_i(a)$ accordingly. Thanks.

---

> > ### Comment · Reviewer_82fy · 2025-04-04
> >
> > I thank the authors for their response and addressing my concerns. I will maintain my positive score.

---

### Official Review · Reviewer_AxPZ · 2025-03-12

**Overall Recommendation:** 2

**Summary:**

This paper introduces a general framework, referred to as SIMULATION, that adapts stochastic (i.i.d.) learning algorithms to the random order model without significantly altering their finite-time performance guarantees. The core idea is straightforward: partition the time horizon into blocks of geometrically increasing length, sample an i.i.d. instance from past observations, and use it to train the algorithm in the current block.

**Claims And Evidence:**

The findings are interesting but somewhat expected, given that the random order model is statistically indistinguishable from the i.i.d. model in the asymptotic sense. Additionally, there is a well-established body of literature on best-of-both-worlds algorithms that achieve optimal performance in both stochastic and adversarial settings [3]. Since the random order model lies between these two extremes in terms of generality, do such algorithms not already address the problem in this setting?

A key concern is the omission of highly relevant work [1], which presents near-optimal algorithms for various problems (including online packing, online learning, and feasibility) in both i.i.d. and random order models. Furthermore, reference [2] is also pertinent to this study.

References:

[1] Agrawal, S., & Devanur, N. R. (2014). Fast algorithms for online stochastic convex programming. In Proceedings of the Twenty-Sixth Annual ACM-SIAM Symposium on Discrete Algorithms, pp. 1405-1424.

[2] Devanur, Nikhil R., & Hayes, T. P. (2009). The AdWords problem: Online keyword matching with budgeted bidders under random permutations. In Proceedings of the 10th ACM Conference on Electronic Commerce, pp. 71-78.

[3] Bubeck, S., & Slivkins, A. (2012). The best of both worlds: Stochastic and adversarial bandits. In Conference on Learning Theory, pp. 42-1. JMLR Workshop and Conference Proceedings.

**Essential References Not Discussed:**

[1] Agrawal, S., & Devanur, N. R. (2014). Fast algorithms for online stochastic convex programming. In Proceedings of the Twenty-Sixth Annual ACM-SIAM Symposium on Discrete Algorithms, pp. 1405-1424.

[2] Devanur, Nikhil R., & Hayes, T. P. (2009). The AdWords problem: Online keyword matching with budgeted bidders under random permutations. In Proceedings of the 10th ACM Conference on Electronic Commerce, pp. 71-78.

[3] Bubeck, S., & Slivkins, A. (2012). The best of both worlds: Stochastic and adversarial bandits. In Conference on Learning Theory, pp. 42-1. JMLR Workshop and Conference Proceedings.

**Experimental Designs Or Analyses:**

No experimental result has been reported.

**Methods And Evaluation Criteria:**

This seems to be fine.

**Other Comments Or Suggestions:**

Typographical Error:
•	Line 155, Second Column: "this can be tough ..."

**Other Strengths And Weaknesses:**

N/A

**Questions For Authors:**

1.	What happens when adversarial algorithms are applied in the random order setting? Do they yield improved guarantees in this model?
2.	Is the proposed transformation universal in the sense that it provides a best-of-both-worlds result for both random order and i.i.d. inputs? Given the extensive literature on best-of-both-worlds models achieving optimal performance in stochastic and i.i.d. settings [3], the authors should discuss how their work relates to this body of research.
3.	How do the results in this paper compare to those in [1] and [2]?
4.	The counterexample presented in Section [3] appears somewhat artificial. What happens if one applies a standard no-regret algorithm, such as UCB, to the random order input?
5.	Theorem 4.5 is meaningful only when $B=O(T)$. If $\rho$ is small, the bound loses significance. This contrasts sharply with the results of Immorlica et al. (2022), which achieve an optimal competitive ratio even for small budgets.

References:

[1] Agrawal, S., & Devanur, N. R. (2014). Fast algorithms for online stochastic convex programming. In Proceedings of the Twenty-Sixth Annual ACM-SIAM Symposium on Discrete Algorithms, pp. 1405-1424.

[2] Devanur, Nikhil R., & Hayes, T. P. (2009). The AdWords problem: Online keyword matching with budgeted bidders under random permutations. In Proceedings of the 10th ACM Conference on Electronic Commerce, pp. 71-78.

[3] Bubeck, S., & Slivkins, A. (2012). The best of both worlds: Stochastic and adversarial bandits. In Conference on Learning Theory, pp. 42-1. JMLR Workshop and Conference Proceedings.

**Relation To Broader Scientific Literature:**

Please see the comments above.

**Theoretical Claims:**

The proofs are clear.

---

> ### Author Rebuttal · Authors · 2025-04-01
>
> We thank the reviewer for their feedback about the paper.
>
> **Adversarial algorithms in RO** As we mention in Section 1.1., and detail in Appendix B, there is a natural hierarchy between the i.i.d., random order, and adversarial input models. In particular, any learning algorithm for the adversarial setting retains its regret bounds against RO inputs. However, an algorithm with optimal regret bounds in the adversarial setting may not be optimal in the (easier) RO model. Consider, for example, the bandits with switching cost problem, where the adversarial minimax regret is $\Theta(T^{2/3})$, as opposed to the stochastic (and RO) rate of $\Theta(\sqrt{T})$. If we were to use the optimal (in the adversarial setting) algorithm by Arora, et al. “Online bandit learning against an adaptive adversary: from regret to policy regret” (ICML 12) on a RO instance, we would obtain a (suboptimal) $\Theta(T^{2/3})$ regret bound (note, in fact, that their approach consists in dividing the time horizon in $T^{2/3}$ time batch, so the algorithm switches freely action between consecutive time batches, for a switching budget $\Omega(T^{2/3})$). A similar argument can be carried out for the prediction with delays model, where adversarial algorithms still retain their multiplicative dependence on the delay parameter, regardless of the input structure.
>
> **Comparison with Best-of-both-worlds literature [3]** The reviewer asks “Since the random order model lies between these two extremes [adversarial and i.i.d.] in terms of generality, do such algorithms [BoBW algorithms] not already address the problem in this setting?”. The answer to this question is, in general, negative. In the typical BoBW literature (as e.g., [3]), the goal is to design learning algorithms with good instance-independent bounds in the adversarial setting and logarithmic instance-dependent bounds on stochastic instances. An algorithm with these properties is only guaranteed to retain its adversarial instance-independent bounds on the random order instance, not its instance-dependent stochastic guarantees; this implies its suboptimality as soon as there is a gap between adversarial and random order. On the contrary, our paper provides a general template to construct tight algorithms for the RO model that match the minimax regret for the i.i.d. scenario.
>
> **Comparison with other related works** We thank the reviewer for the suggested literature. We will incorporate a discussion of both references in the final version of the paper. Although close in spirit to our research agenda of investigating the relationship between random order and i.i.d. inputs in online algorithms, our model and results are orthogonal to the ones studied in [1,2]. Consider, for instance, our online learning with constraints model. At each time step, our learning algorithm selects an action before observing the losses and constraint violations. In contrast, the online algorithm for Online Stochastic Convex Programming [1] first observes the losses and violations, and only then makes a decision. In other words, our setting falls within the domain of online learning, whereas theirs falls within the domain of competitive analysis.
>
> **Counterexample** Regarding the point raised on the counterexample, the primary goal behind our construction is to provide a general framework that extends the applicability of algorithms designed for the i.i.d. setting beyond purely stochastic environments. Even though UCB may work “as-is” in this specific random order instance, the counterexample shows that not all algorithms that achieve sublinear regret bounds in the i.i.d. model automatically have sublinear bounds on regret in the random order model. This highlights the need for our general construction. We hope that the power of this general template is evident from the various instantiations we present across different online problems.
>
> **Results of Immorlica et al** We are not sure we understand the concern raised by the reviewer regarding Immorlica et al. Indeed, the competitive ratio appears only in the adversarial setting of Immorlica et al. while here we show no-regret (ie, competitive ratio = 1). Our results align with those established by previous work for the stochastic setting. Specifically, we refer the reviewer to the stochastic regret guarantees in the work of Immorlica et al. There, they show a regret bound of $T/B\cdot \sqrt{T}$, which is $\sqrt{T}/\rho$ in our notation. Note that if we employed the algorithm of Immorlica et al. as a subroutine, we would still obtain the same regret rates as in Corollary 4.8. Moreover, in all cases in which $T\sqrt{T}/B$ is meaningful (i.e., when $B \ge \Omega(\sqrt{T})$), we obtain the same exact rate of $T/B\sqrt{T}$ (since the term $1/\rho^2$ becomes negligible as $1/\rho^2=(T/B)^2 \le T\sqrt{T}/B$).
>
> If any concerns remain on this matter, we are happy to clarify further. Otherwise, we encourage the reviewer to reconsider this point in light of our answer.

---

### Official Review · Reviewer_iXLi · 2025-03-17

**Overall Recommendation:** 4

**Summary:**

The paper studies a general online learning setting where in every round $t \in [T]$ there is an unknown loss vector $\ell_t$, the learner needs to make a decision $x_t \in \mathcal{X}_t$, and incurs loss $\langle x_t, \ell_t\rangle$ (there might also be constraints that need to hold across the whole time horizon, like budget constraints.) Unlike the fully adversarial setting, the authors consider the case where the loss vectors are chosen by the adversary and then a random permutation of them is presented to the algorithm. The authors show that there exist (pathological) stochastic algorithms that do not work in this model. Then, the authors propose a template that can be used to obtain a "stochastic-to-online" transformation in their setting which works by breaking the interaction into logarithmic many intervals of doubling size, and using the interval $t-1$ to train the stochastic algorithm and apply it appropriately to interval $t$. They instantiate the template in various setting including online learning with delayed feedback, online learning with budget constraints, and Littlestone's online learning. The key technical observation is that the performance of the stochastic algorithm within consecutive intervals should be very close (e.g., Claim 4.3).

**Claims And Evidence:**

Yes, they are supported by proofs.

**Essential References Not Discussed:**

References discussed.

**Experimental Designs Or Analyses:**

N/A.

**Methods And Evaluation Criteria:**

The claims contain proofs, which make sense.

**Other Comments Or Suggestions:**

- “The empirical distribution from this phase is then applied to the actual instance within the current block.” -> not sure what this means, do you mean the model trained on i.i.d. Samples from the empirical distribution of this phase?

- Why is “follow-the-leader” no-regret in the adversarial setting? Maybe follow the perturbed leader? (follow the leader suffers from switching the chosen action all the time and being one step behind from the optimal algorithm).


- “Surprisingly, there exists a random-order instance that fools BIRTHDAY-TEST.” -> I think the birthday-test algorithm is interesting, but I don’t think it’s surprising that it is fooled by a random-order input; given the construction (which is nice), I think it is clear that this algorithm will get confused.

- I wouldn’t call it “procedure” because this reads more as a black-box transformation but rather a “template”, but this is just personal preference.

**Other Strengths And Weaknesses:**

Strengths:

- The authors consider a pretty general online learning setting and study a beyond-worst-case analysis of it by considering random permutations of the input sequence.

- The authors propose a pretty general template and they instantiate it in various settings showing strong results.

- The technical ideas are solid (at least they are above the bar of ICML).

Weaknesses:

- I don't see any strong weaknesses. Personally, I believe the claim that the paper "initiates the systematic study of the random- order input model in online learning." is a bit of stretch. I agree that the setting the authors study is more general than prior work, but special cases of it have been studied extensively (e.g. OCO with random permutation arrivals).

**Questions For Authors:**

No further questions, please look at the comments.

**Relation To Broader Scientific Literature:**

The results are of interest to theorists mostly, and practitioners secondarily.

**Theoretical Claims:**

The claims seem to be sound.

---

> ### Author Rebuttal · Authors · 2025-04-01
>
> We thank the reviewer for their detailed comments and positive feedback about the paper.
>
> Regarding the specific comments and suggestions proposed:
> 1. Yes, that is correct. We will rephrase the sentence to make it clearer.
> 2. As correctly stated by the reviewer, Follow-the-leader is no-regret only in the stochastic setting and may perform poorly in the adversarial setting. We use Follow-the-leader as a subroutine in BIRTHDAY-TEST, where we only need its no-regret property on stochastic inputs. Indeed, we only claim that BIRTHDAY-TEST is no-regret in the stochastic setting, as any no-regret algorithm for the adversarial setting automatically retains its no-regret property on random order instances.
> 3. And 4. Thanks for the suggestions; we will rephrase such comments and update the wording accordingly.

---

### Decision · Program_Chairs · 2025-05-01

**Decision:**

Accept (poster)

**Comment:**

This work studies online learning in the random-order model, an interesting middle ground (in the finite case) between the i.i.d. stochastic model and adversarial orders. In particular, the authors:
1. show that directly applying a no-regret algorithm for the stochastic model can fail against a random-order instance (i.e., they demonstrate a separation between the two models), and
2. design a sublinear-regret algorithm called `Simulation` that uses a low-regret algorithm in the stochastic model as a black-box subroutine.

Overall, the reviewers lean towards acceptance (scores: [4, 2, 3, 3]). Note that Reviewer AxPZ (score: 2) did not respond to the authors' rebuttal. The paper is very well written, and in general we appreciate bringing more attention to beyond-worst-case bounds.